# Conserved signalling components coordinate epidermal patterning and cuticle deposition in barley

Linsan Liu[1,6], Sarah B. Jose [2,6], Chiara Campoli[1], Micha M. Bayer [3], Miguel A. Sánchez-Diaz [1], Trisha McAllister [1], Yichun Zhou[1], Mhmoud Eskan[1], Linda Milne[3], Miriam Schreiber[3], Thomas Batstone[2], Ian D. Bull [4], Luke Ramsay [3], Penny von Wettstein-Knowles [5], Robbie Waugh [1,3], Alistair M. Hetherington [2] & Sarah M. McKim [1] ✉

Faced with terrestrial threats, land plants seal their aerial surfaces with a lipid-rich cuticle. To breathe, plants interrupt their cuticles with adjustable epidermal pores, called stomata, that regulate gas exchange, and develop other specialised epidermal cells such as defensive hairs. Mechanisms coordinating epidermal features remain poorly understood. Addressing this, we studied two loci whose allelic variation causes both cuticular wax-deficiency and misarranged stomata in barley, identifying the underlying genes, *Cer-g/ HvYDA1*, encoding a YODA-like (YDA) MAPKKK, and *Cer-s/ HvBRX-Solo*, encoding a single BREVIS-RADIX (BRX) domain protein. Both genes control cuticular integrity, the spacing and identity of epidermal cells, and barley's distinctive epicuticular wax blooms, as well as stomatal patterning in elevated $CO_2$ conditions. Genetic analyses revealed epistatic and modifying relationships between *HvYDA1* and *HvBRX-Solo*, intimating that their products participate in interacting pathway(s) linking epidermal patterning with cuticular properties in barley. This may represent a mechanism for coordinating multiple adaptive features of the land plant epidermis in a cultivated cereal.

Plants living on land face many threats, such as desiccation, UV radiation, gravity, pests and temperature extremes. To protect themselves, land plants seal their aerial surfaces with a hydrophobic and reflective cuticle and develop specialised epidermal cells such as defensive hairs[1–3]. Land plants interrupt their cuticle with epidermal pores called stomata that balance photosynthetic gas exchange with transpiration[4,5]. Further epidermal elaboration helped plants colonise new terrestrial niches; for instance, the fast-responding four-cell stomata complexes and thick, waxy cuticles of grasses likely contributed to their expansion during global aridification 35–40 million years ago, and impact current cereal yields[6–9]. Despite the adaptive and agronomic importance of

epidermal features, whether their development is co-ordinated has been little explored.

The pathways contributing to epidermal cell identity and distribution are better studied. The outer epidermis of most monocot leaves and stems consists of parallel axial files of elongated pavement cells. However, depending on the lineage, files can also contain specialised cell types, including stomatal complex cells, silica–cork cell pairs and epidermal hairs[10–12]. All epidermal files originate from protodermal cells, which acquire their identities along an acropetal gradient in the growing leaf from base to tip[13]. In grasses, stomatal precursor cells each undergo a single asymmetric cell division (ACD) to generate a larger basal daughter cell and a smaller apical cell. The

[1]Division of Plant Sciences, School of Life Sciences, University of Dundee, Dundee, UK. [2]School of Biological Sciences, University of Bristol, Bristol, UK. [3]James Hutton Institute, Dundee, UK. [4]School of Chemistry, University of Bristol, Bristol, UK. [5]Department of Biology, University of Copenhagen, Copenhagen, Denmark. [6]These authors contributed equally: Linsan Liu, Sarah B. Jose. ✉e-mail: smckim@dundee.ac.uk

larger cell differentiates into a pavement cell and the smaller cell adopts the guard mother cell identity. The guard mother cell subsequently recruits two subsidiary cells from neighbouring files and divides symmetrically into two dumbbell-shaped guard cells, making the four-cell stomatal complex[14]. Thus, pavement cells alternate one to one with stomatal complexes, an arrangement that ensures stomata do not interfere with each other[5,15]. Although not arranged in files, dicots and some monocots also distribute their stomata according to the "one-cell spacing" rule. Best understood in Arabidopsis, the molecular spacing mechanism involves restricting the activity of stomatal promoting factors to the smaller daughter cell, including a core set of basic helix-loop-helix (bHLH) transcription factors, SPEECHLESS (SPCH), MUTE, and FAMA, which work together with either INDUCER OF CAB EXPRESSION1/SCREAM (ICE/SCRM) or SCRM2 to promote stomatal identity, guard cell division and termination, respectively[3]. A mitogen-activated protein kinase (MAPK) cascade initiated by a Ste11-class MAP kinase kinase kinase (MAPKKK) called YODA (YDA), and mediated downstream by MKK4/5 and MPK3/6[16], inhibits stomatal bHLHs in the larger cell. YDA function becomes polarised towards the larger daughter cell during ACD by the intrinsic polarity protein BREAKING OF ASYMMETRY IN THE STOMATAL LINEAGE (BASL), which assembles YDA, along with other signalling proteins, into a polarity complex on the stomatal precursor cell periphery inherited by the larger daughter cell[17–19]. Polar localisation of BASL depends on its interactions with BREVIS RADIX (BRX) and POLAR scaffolding proteins[20,21], and the BRX-domain containing PRAF (plextrin, regulator of chromatin condensation and FYVE domain) proteins[22]. Stomatal development in grasses is not as clearly elucidated but appears to share many core bHLH and signalling elements, although with modified circuitry[14]. For instance, in Brachypodium, rice and maize, MUTE laterally moves from the guard mother cell to recruit subsidiary cells[23] while in Brachypodium BdYDA1 ensures one-cell spacing by reinforcing asymmetric cell fate following, but not before, the ACD[24]. Indeed, BdYDA1 is not preferentially inherited into the larger cell, instead showing a broad expression in both daughter cells after ACD; an observation linked to a lack of BASL genes outside of the eudicots and the hypothesis that grasses use alternative mechanisms[24,25].

In addition to acquiring distinct cell identities, the outer epidermal layer forms an overlying cuticle during organ development. The cuticle is composed of waxes embedded in a crosslinked cutin matrix covered with epicuticular surface waxes arranged as films and/or crystals[1]. Basic biosynthetic pathways for the ubiquitous epicuticular waxes were established biochemically[26], but have been greatly extended by exploiting eceriferum (cer, 'not bearing wax') and glossy mutants in several species[2]. Cuticular waxes originate from plastid-synthesised $C_{16}$ or $C_{18}$ fatty acids elongated by β-ketoacyl-CoA synthases (KCSs) in the endoplasmic reticulum into very long chain fatty acids (VLCFAs, ≥ 20 carbons), which are transported through the plasma membrane and onto the plant surface[1]. However, other VLC aliphatics are found often in dominating amounts. For instance, graminoid cereals like barley and wheat, deposit epicuticular $C_{31}$ β-diketone and $C_{31}$ hydroxy-β-diketone crystalline tubes on reproductive tissues, including emerged leaf sheaths, exposed internodes and spike inflorescences, causing a blue-grey glaucous 'wax bloom', associated with drought tolerance, reflectance and enhanced yields[27–29]. Cloning the genes underlying barley cer-cqu mutants revealed that β-diketone biosynthesis (DKS) depends on the activity of a novel diketide synthase (DKS) early in elongation of the carbon chains compared to only fatty acid synthases as occurs during the formation of the ubiquitous aliphatics[30,31]; however, we know little about upstream regulation of β-diketone biosynthesis and deposition[28].

More broadly, how plants coordinate epidermal development with cuticle formation is a pressing question. Careful study of epidermal fate in wax-deficient mutants and cuticular properties of epidermal patterning mutants suggests that these aspects of development could be intertwined. For instance, several Arabidopsis mutants defective in wax biosynthesis show altered stomatal indexes, including in response to elevated $CO_2$[32,33], while the overexpression of SHINE1/WAX-INDUCER1 (SHN1/WIN1) transcription factors in both Arabidopsis and tomato influence trichome and stomatal development in addition to cuticular wax deposition[34,35]. Interestingly, the glossy barley cer-g mutants violate the "one-cell spacing rule", producing clustered, directly appressing stomata, leading to the suggestion that Cer-g could encode an enzyme common to wax biosynthesis and stomatal development[36].

Here, we show that cer-g mutants have weakened cuticular integrity and clustering of other epidermal cell types. We also discovered that barley cer-s (glossy sheath5) mutants[37], show similar but typically more severe phenotypes than cer-g. We reveal that the cer-g and cer-s mutants arise from defective alleles in barley YDA and BRX-domain-containing proteins, respectively, suggesting that BRX-domain proteins also control epidermal spacing in grasses and that both YDA and BRX-domain factors control multiple epidermal specialisations important for terrestrial survival and cereal crop performance.

## Results
### Variations in stomatal signalling pathway components underlie cuticular mutant phenotypes in barley

To explore the genetic control of cuticle formation in grasses, we examined the cer-g.10 and cer-s.31 mutants described as having discontinuous wax deposition. We confirmed that these mutants have a grainy (patchy) distribution of wax on their uppermost leaf sheaths, internodes, and spikelet organs such as lemmas, compared with the evenly distributed glaucous wax observed in their wild-type parent cultivar (cv.) Bonus (Fig. 1a; Supplementary Fig. 1a); cer-s.31 spike organs appeared much glossier compared to cer-g.10. Scanning electron microscopy revealed a dense mat of long, thin, crystalline tubes on the leaf sheath of Bonus and a patchy distribution of low- and high-density wax tubes on the mutants (Fig. 1a). The abaxial surface of the first leaf of the wild type and each of the single mutants showed no obvious difference in wax crystals (Supplementary Fig. 1b). Mutant and Bonus leaves, spikes or leaf sheaths showed no difference in staining when immersed in toluidine blue for 90 min[38], but longer incubations of spikes for 5 h, leaves for 24 h and leaf sheaths for 28 h revealed increased staining in the mutants, with more intense staining in spikes and leaves of cer-s.31 compared to cer-g.10 (Supplementary Fig. 1c). These results suggest that cer-g.10 and cer-s.31 impair cuticular integrity in both tissues synthesising (spikes and leaf sheaths), and those lacking (leaves) β-diketone aliphatics.

Consistent with the grainy wax distribution, partial compositional analysis using gas chromatography-mass spectrometry (GC–MS) showed that cer-s.31 spikes produced 42% fewer $C_{31}$ β-diketones and 33% fewer $C_{31}$ hydroxy-β-diketones, respectively, compared with Bonus ($P < 0.05$, Supplementary Fig. 2a; Supplementary Data 1). The total amount of leaf wax and its composition showed little difference across the genotypes ($P < 0.05$; Supplementary Fig. 2b; Supplementary Data 1), as expected given the similarity in leaf wax density and crystal structure (Supplementary Fig. 1b). Examining the large allelic series for the cer-g and cer-s mutants (Supplementary Data 2) showed that most cer-s lines had little visible wax on their flag leaf sheaths and internodes, while the cer-g mutants generally produced patchy wax on these organs, consistent with the larger reduction of $C_{31}$ β-diketone aliphatics on cer-s.31 spikes compared with cer-g.10 (Supplementary Data 1). Reduced internode elongation in cer-g.10 and cer-s.31 led to 10% and 8% height reductions compared to Bonus ($P < 0.05$), respectively, while root lengths were equivalent (Supplementary Fig. 3). Thus, variations at Cer-g and Cer-s play major roles regulating β-diketone epicuticular wax deposition, with minor roles in other traits such as cuticular integrity and internode length.

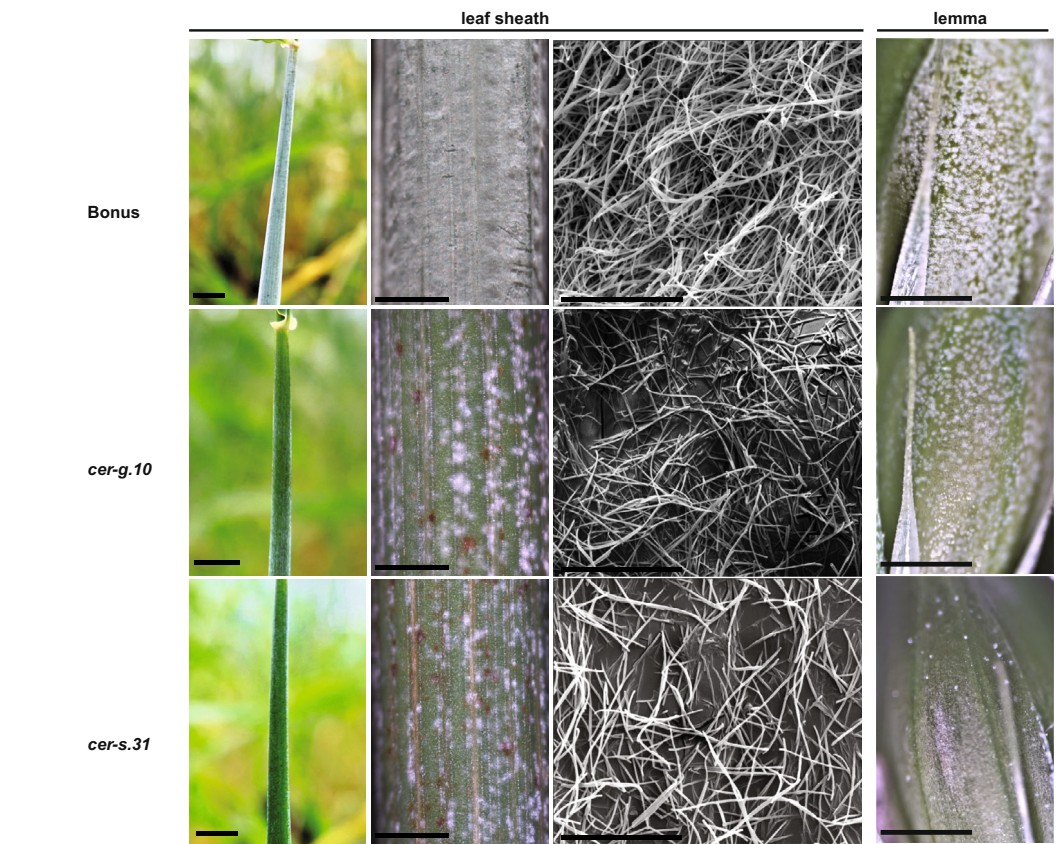

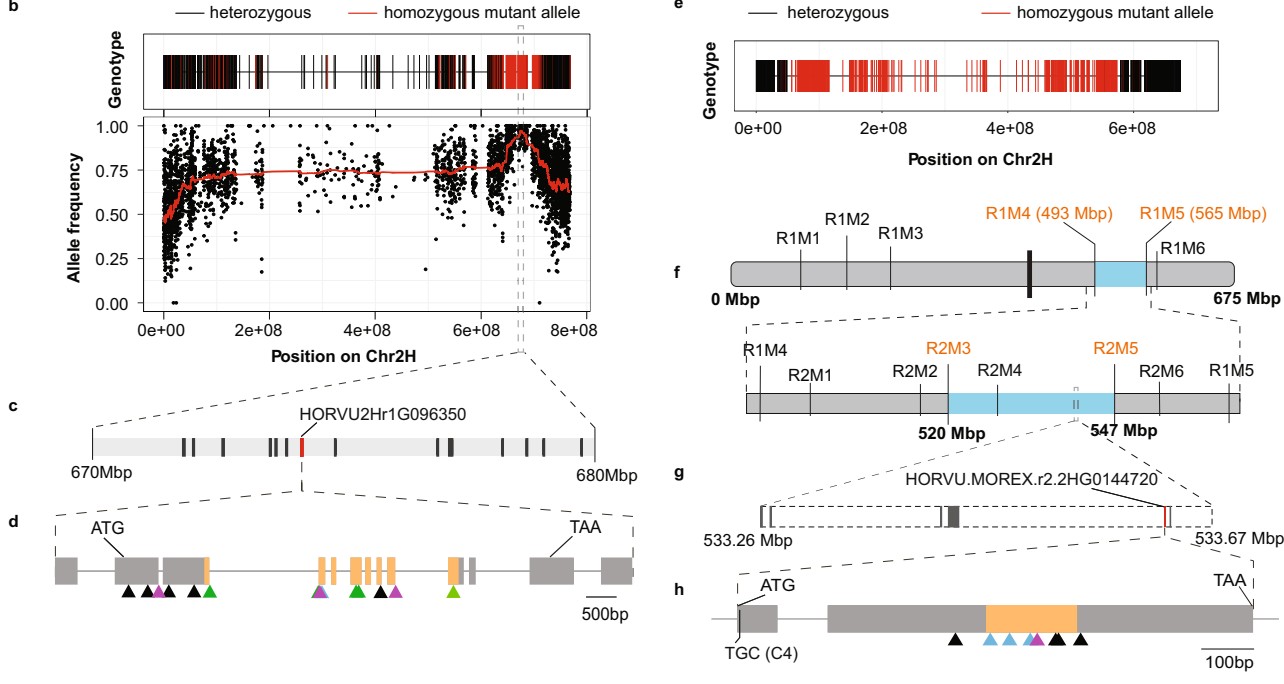

▲ stop codon   ▲ amino acid alteration   ▲ amino acid deletion/insertion   ▲ frame shift

To identify the underlying genes, we exploited Bowman near-isogenic lines (BwNILs), BW111 and BW122, generated by repeatedly back-crossing *cer-g.10* and *cer-s.31*, respectively, to cv. Bowman[39] which show similar wax phenotypes to the original mutants (Supplementary Fig. 4). Consistent with previous mapping to chromosome 2H[37,40], genotyping using the Barley 50k iSelect SNP Array[41] identified four Bonus introgressions on chromosome 2H in BW111 and one introgression on chromosome 2H and one more on chromosome 5H in BW122 (Supplementary Fig. 5; Supplementary Data 3). SNP genotyping of a BW111 × cv. Morex F$_2$ population phenotypic bulk followed by exome capture sequencing revealed an elevated mutant allele variation frequency from 640 Mbp to 720 Mbp on chromosome 2H on the

**Fig. 1 | Variation in signalling genes causes defects in epicuticular wax accumulation in barley. a** Epicuticular wax on the leaf sheaths and lemmas of cv. Bonus, *cer-g.10* and *cer-s.31*. Panels from far left: leaf sheath (scale bars = 1 cm), close-ups of sheaths (scale bars = 1mm), scanning electron micrographs of sheaths (10,000× magnification, scale bar = 10 μm; N = 4 per genotype), lemmas (scale bars = 2 mm). **b**–**d** Cloning of *cer-g* using bulked segregant analysis (BSA) coupled to Barley 50k iSelect SNP genotyping (BSA-50 K) and exome capture (EC) sequencing (BSA-ECseq) based on the Morex V1 assembly[95]. **b** BSA-50 K (upper) and BSA-ECseq (below) from a BW111 × Morex F2 phenotypic bulk. Black dots represent the frequency of the mutant alleles on chromosome 2H (Chr2H). Average allele frequencies shown by the red line define a candidate region around 675 Mbp. **c** Close up of the 670–680 Mbp region. Bars represent the genomic regions of candidate genes associated with epidermal features. The red bar indicates *HORVU2Hr1G096350* (*HvYODA1*, *HvYDA1*) which has a 3-bp deletion in BW111 mutants compared with the parental cultivars. **d** *HvYDA1* gene model. Bars represent exons and regions encoding the kinase domain are coloured orange. Allelic *cer-g*

variants are indicated underneath using triangles. **g**, **h** Cloning of *cer-s* using BSA-50 K followed by fine mapping and whole-genome sequencing based on Morex V2 assembly[42]. **e** BSA-50 K conducted on BW122 x Morex F2 phenotypic bulk located the causal mutation on Chr2H. **f** Fine mapping using markers indicated above, narrowed the candidate region down to 520 – 547 Mbp on Chr2H. **g** Whole-genome sequencing identified a BW122-specific deletion from 533.26 – 533.67 Mbp relative to the parent cultivars. Bars indicate genes within the deletion boundaries. The red bar indicates *HORVU.MOREX.r2.2HG0144720* (*HvBREVIS-RADIX-Solo*, *HvBRX-Solo*), which co-located with SNPs identified in the *cer-s* alleles. **h** *HvBRX-Solo* gene model. Bars represent exons; the region encoding the BRX domain is coloured orange. A predicted S-acylation site (C4) is indicated below. Variants identified in *cer-s* alleles are indicated underneath in triangles. Triangle colours indicate different variant types: stop codon variants shown in black; amino acid substitution shown in blue; amino acid insertions/deletions shown in green; and frame–shift variants shown in red. Source data of **b** are provided in the Source data file.

Morex V1 assembly, with a single peak at ~675 Mbp (Fig. 1b). Filtering between 670–680 Mbp for high-confidence (HC) genes with ontologies related to epidermal features resolved 16 genes, one of which encoded a YDA-like MAPKKK (*HORVU2Hr1G096350*, *HvYDA1*) that had a BW111-specific three base-pair deletion in the predicted kinase domain, replacing the serine at 507 and tyrosine at 508 with an asparagine (Fig. 1c,d; Supplementary Data 2,4). The original *cer-g.10* allele had the same deletion. Sequencing the *cer-g* allelic mutants identified 16 *HvYDA1* alleles presenting 14 independent variants that disrupt the predicted kinase domain (Fig. 1d; Supplementary Data 2), with 20 alleles showing a potential complete or partial gene deletion (Supplementary Data 2), confirming *HvYDA1* (*HORVU.MOREX.r2.2HG0155920* in Morex V2[42] and *HORVU.MOREX.r3.2HG0188270* in Morex V3[43]) as *Cer-g*.

To identify *Cer-s*, we similarly genotyped a BW122 × Morex F2 phenotypic bulk, revealing a candidate region spanning much of chromosome 2H (Fig. 1e). Fine mapping using KASP markers on 313 F2 individuals narrowed this to a 72 Mbp interval from 493–565 Mbp on the Morex V2 assembly, and further mapping of 45 recombinants with new markers narrowed the interval to a 27 Mbp region from 520–547 Mbp containing 248 HC genes on the Morex V2 assembly[42] (Fig. 1f). In parallel, we mapped whole-genome sequencing reads from BW122, Bowman and Bonus to the Morex V2 genome, followed by small (<50 bp) and structural (> 8 bp) variant calling. After a manual review of the raw mapping data, we confirmed a BW122-specific deletion at 533.26–533.67 Mbp on chromosome 2H (Fig. 1g). We could not amplify the six HC genes within this region (Supplementary Data 4) from BW122 or *cer-s.31*. One of these genes, *HORVU.MOREX.r2.2HG0144720*, encodes a BRX family (BRXf) protein containing a solitary BRX domain. This gene, which we name *HvBRX-Solo*, showed variation predicted to impair its function in all glossy *cer-s* alleles (Fig. 1h; Supplementary Data 2). Most *cer-s* alleles alter the BRX domain, while *cer-s.1153* has a truncated downstream C-terminal region, suggesting that both the single BRX domain and the C-terminal region contribute to HvBRX-Solo function. Three *cer-s* alleles − *cer-s.1054*, *cer-s.24* and *cer-s.372* − did not show glossy phenotypes or changes in *HvBRX-Solo* sequence. F1 progeny from each of these alleles crossed to *cer-s.31* showed waxy phenotypes (Supplementary Fig. 7a), complementing the glossy *cer-s.31* phenotype, suggesting that *cer-s.1054*, *cer-s.24* and *cer-s.372* are not *cer-s* alleles. Taken together, our data support the conclusion that *HvBRX-Solo* (*HORVU.MOREX.r3.2HG0175040* in Morex V3) is *Cer-s*.

Using RTqPCR, we detected *HvYDA1* and *HvBRX-Solo* transcripts in young leaves, developing sheaths and spikes in cv. Bowman (Supplementary Fig. 6a, b, e). Evidence from the EORNA barley transcript abundance database[44], suggests that another YDA-like gene, *HvYDA2* (HORVU6Hr1G064150; HORVU.MOREX.r3.6HG0602230 in Morex V3), is similarly expressed as *HvYDA1* across 16 vegetative and reproductive

tissues in cv. Morex, at between 44 and 118% *HvYDA1* levels, depending on the tissue (Supplementary Fig. 6c, d). We also examined the expression of *HvYDA1* and *HvBRX-Solo* by RTqPCR in five sections of developing flag leaf sheath with section 1 at the bottom and section 5 just after emergence from the enclosing leaf sheath. *HvBRX-Solo* and *HvYDA1* were exclusively and preferentially expressed in the younger sheath sections, respectively, tissues likely undergoing division, patterning decisions and cuticle formation[45] (Supplementary Fig. 6e). In contrast, transcripts from *Cer-CQU* gene cluster involved in the biosynthesis of the epicuticular wax bloom, showed low abundance in young sections and elevated levels in tissues just prior to emergence (Supplementary Fig. 6e). Collectively, our data reveal that *HvYDA1* and *HvBRX-Solo* promote cuticular integrity and wax deposition in barley, with expression dynamics suggesting an early role for *HvYDA1* and *HvBRX-Solo* during sheath development and separated from later metabolic gene expression associated with epicuticular wax deposition.

### *HvYDA1* and *HvBRX-Solo* promote robust epidermal patterning in barley

Initial reports of *cer-g.10* described abnormal stomatal clustering on leaf blades and other organs[36,46], aligning well with roles described for YDA in Arabidopsis and Brachypodium[24,47]. We confirmed this phenotype and discovered similar phenotypes in *cer-s.31*. Stomata, as well as prickle hair cell or a silica–cork cell pairs, also show one-cell spacing with pavement cells originating from asymmetric cell fate following an ACD (Fig. 2a). Similar to *Bddya1*[24], *cer-g.10* and *cer-s.31* mutants also showed clustering of prickle hair cells or a silica–cork cell pairs. Here we quantitatively describe stomatal phenotypes of *cer-g.10* and *cer-s.31*, then the prickle hair and silca-cork cell pair phenotypes.

The abaxial surfaces of the first leaf of 14-day (d)-old Bonus showed that stomatal cell files follow the one-cell spacing rule, producing double stomatal clusters in only 0.1% of stomatal events (both a normal stoma or a stomatal cluster were counted as one event). On the other hand, 5.0% *cer-g.10* and 2.3% *cer-s.31* stomatal events occurred as stomatal clusters ($P < 0.05$), mostly comprised of double or triple clusters with infrequent quadruple clusters (Fig. 2b, c; Supplementary Fig. 8a; Supplementary Data 5). Compared with 1.6% of Bonus stomatal complexes, 6.4% *cer-g.10* and 5.1% *cer-s.31* stomatal complexes on the abaxial leaf surfaces developed misshaped elongated subsidiary cells, most commonly extending over neighbouring cells to flank two separate guard cell pairs ($P < 0.05$; Fig. 2b,d; Supplementary Fig. 8a; Supplementary Data 5). In addition, 14.8% *cer-g.10* and 10.4% *cer-s.31* abaxial stomata were separated by more than one epidermal cell compared with only 2.9% in Bonus ($P < 0.05$; Fig. 2b, e). These supernumerary epidermal cells were usually smaller and rounder than a typical pavement cell, resembling a guard mother cell that had failed to

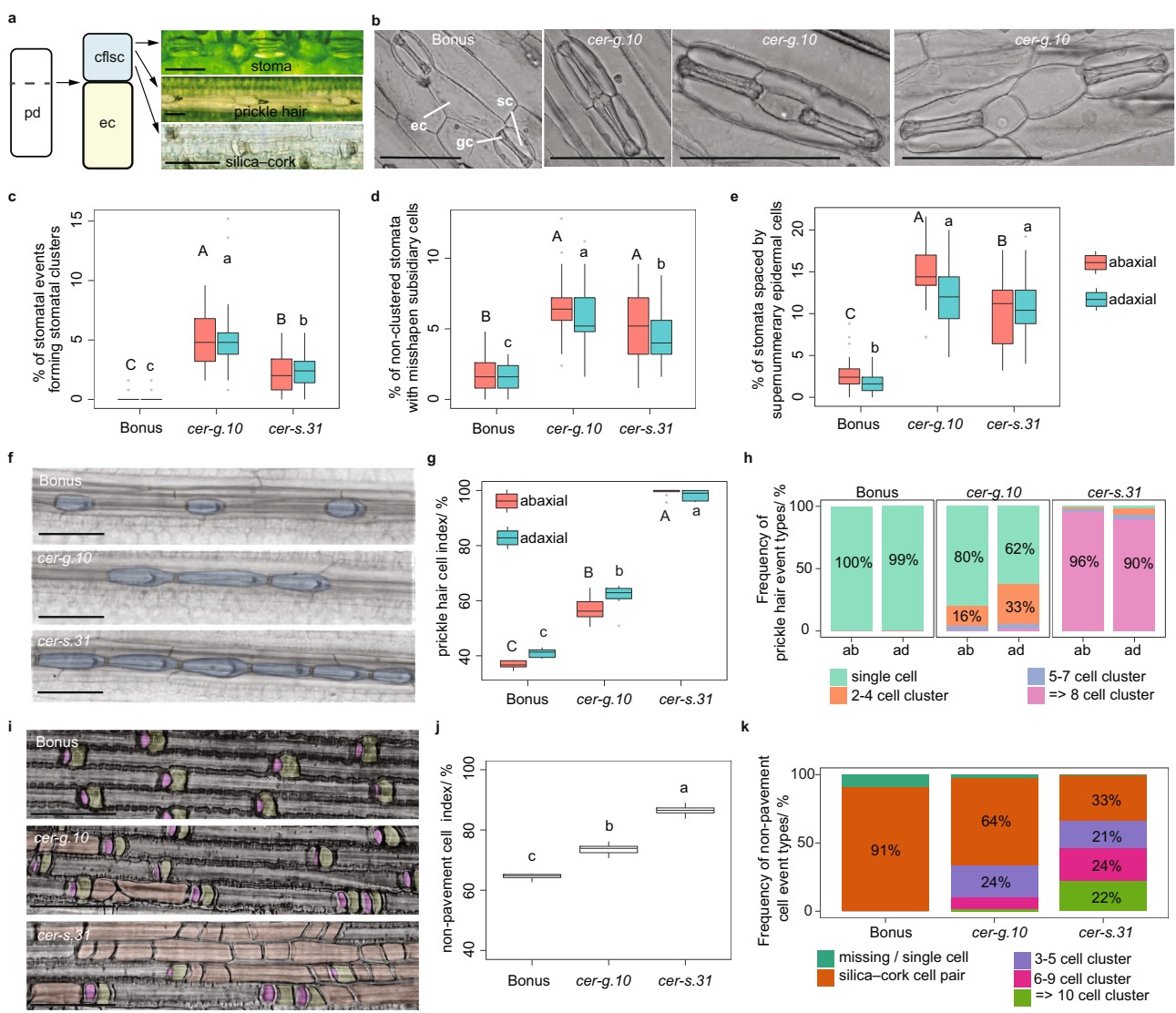

**Fig. 2 | Variation in *HvYDA1* and *HvBRX-Solo* alters epidermal patterning.**
**a** Model of epidermal differentiation. A protodermal cell (pd) undergoes asymmetric cell division. The larger daughter cell develops into an epidermal pavement cell (ec). Depending on file lineage, the smaller cell file lineage-specific cell (cfls) differentiates into either a stoma, silica–cork cell pair or prickle hair cell. Scale bar = 50 μm. *N* = 8 plants per cell file type. **b** The left panel shows a representative image of stomatal patterning in Bonus where stomatal complexes composed of guard cells (gc) and subsidiary cells (sc) are separated by a single epidermal cell. The right panels show representative images from *cer.g.10* of the abnormal patterning observed in *cer.g.10* and *cer.s.31*. From the left: directly appressing, 'clustered' stomatal complexes; stomata with misshapen subsidiary cells; and a subsidiary cell extending over the neighbouring cell with supernumerary cells between the stomata. Scale bars = 100 μm. *N* = 24 plants per cell event type.
**c–e** Percentage of stomatal events that were **c** clustered stomata, **d** non-clustered stomata with misshapen subsidiary cells and **e** pairs of stomata separated by two or more epidermal cells within all stomatal events in the first leaf of 14-d-old Bonus, *cer.g.10* and *cer.s.31*. *N* = 24 individuals per genotype, with 125 stomatal events measured per individual. Different uppercase and lowercase letters indicate a significant difference between genotypes (*p* = 0.05; Dunn's test following a Kruskal-Wallis test in **c**, and Tukey's HSD multiple comparison following one-way ANOVAs in **d** and **e**) on the abaxial and adaxial side, respectively). **f** Representative images of prickle hair cell files. In Bonus, each prickle hair cell is separated by an epidermal pavement cell while *cer.g.10* and *cer.s.31* show adjacent prickle hair cells. Scale bars = 100 μm. *N* = 8 plants per genotype. **g** Prickle hair cell index in Bonus, *cer.g.10*

and cer-s.*31*, showing the ratio of prickle hair cells to total cells in the prickle hair files. Different letters indicate significant difference (*p* = 0.05; Tukey's HSD multiple comparison following one-way ANOVAs with square-root transformation) between genotypes. *N* = 8 plants per genotype. **h** Stacked bar graphs numbers show frequency of single and clustered (two to four, five to seven and greater than eight) prickle cell hair events in Bonus, *cer.g.10* and cer-s.*31*. *N* = 8 plants per genotype. **i** Representative images of silica–cork cell files. Bonus shows each silica (pink)–cork(yellow) cell pair separated by an epidermal pavement cell while *cer.g.10* and *cer.s.31* present clusters of non-pavement cells with less defined silica–cork identity (orange). Scale bars = 50 μm. *N* = 8 plants per genotype. **j** Non-pavement to all cell index in Bonus, *cer.g.10* and *cer.s.31* silica–cork cell files. Different letters indicate significant difference (*p* = 0.05; Tukey's HSD multiple comparison following one-way ANOVAs with square-root transformation) between genotypes. *N* = 8 plants per genotype. **k** Stacked bar graphs showing frequency of normal silica–cork cell pair and abnormal cell patterning (missing/single cell, cluster of three to five, six to nine and greater than ten non-pavement cells) events in costal (over vasculature) files of Bonus, *cer.g.10* and *cer.s.31* leaf sheaths. *N* = 8 plants per genotype. In panel (**c–e**), (**g**) and (**j**) box plots, the lower and upper box edges represent the first and third quartiles, the horizontal lines indicate the median, and the lower and upper whiskers denote the minimal and maximal values within 1.5* interquartile range, respectively, while points indicate outliers beyond this range. Source data including p values of statistic tests of **c–e**, **g**, **h**, **j** and **k** are provided in the Source data file.

divide (Fig. 2b). Guard mother cells in mutants also very rarely divided in the transverse plane, resulting in 'C'-shaped guard cells instead of the typical dumbbell morphology (Supplementary Fig. 8a). The first leaf within the coleoptile of three-day-old germinating seedlings showed adjacent guard mother cells and extra subsidiary cells at the recruitment stage (Supplementary Fig. 8b), consistent with a loss of identity and determinacy in the presumptive pavement cell, which either adopted a guard mother cell fate and recruited subsidiary cells, or a compromised identity. When stomata within clusters were counted individually, a 15% higher abaxial stomatal density (stomata/mm$^2$) and a 6% increase in the abaxial stomatal index (frequency of epidermal cells which are stomata) were observed in cer-g.10 compared with Bonus ($p < 0.05$; Supplementary Fig. 8c, d); however, the mutants showed no difference in stomatal density or stomatal index when the clusters were treated as single stomatal events (Supplementary Fig. 8e, f), suggesting that compromised HvYDA1 or HvBRX-Solo function does not influence entry into the stomatal lineage and/or the decision to undergo ACD. Adaxial surfaces showed similar phenotypes (Fig. 2c–e; Supplementary Fig. 8c–f). Flag leaf sheaths in the mutants also displayed clustered stomata and produced more supernumerary epidermal cells between stomata than Bonus (Supplementary Fig. 8g). These mutant phenotypes collectively suggest that HvYDA1 and HvBRX-Solo reinforce cell fate and control subsidiary cell number to establish one-cell spacing and correct stomatal complex morphology. We also examined whether stomatal development was sensitive to $CO_2$ levels. We found that elevated $CO_2$ rescued misshapen subsidiary cells in the single mutants ($P < 0.05$; Supplementary Fig. 9a, b) but did not change the number of supernumerary epidermal cells and stomatal spacing in any genotype (Supplementary Fig. 9c).

Similar to the Bdyda1 mutant[24], cer-g.10 and cer-s.31 mutants showed clustering of prickle hair and silica–cork cells (Fig. 2f, i), other epidermal cells showing one-cell spacing following ACD[11,24,48]. We noted that prickle hair cell files occurred in costal (over vasculature) epidermal files. We developed a prickle hair cell index to describe the proportion of the prickle hair cells relative to total cells in the file. The second leaf of 14-d-old plants showed that compared to Bonus, cer-g.10 had 58% and 51% increased prickle hair cell indexes on abaxial and adaxial costal files, respectively, while almost all cells within the cer-s.31 prickle hair files had prickle hair cell identity ($P < 0.05$; Fig. 2f, g). Clusters in cer-s.31 contained more cells than cer-g.10 (Fig. 2h). In grasses, ACD-generated silica mother cells symmetrically divide into a short cell pair, which develop into an upper silica deposition cell and a basal cupping cork cell[48]. We noted that the costal silica–cork cell files of the abaxial mutant leaf sheaths contained cell clusters that were rounder and smaller than a typical pavement cell, potentially representing silica mother cells which failed to divide (Fig. 2i). To represent all cells, we used cell index to describe the proportion of non-pavement cell events (including silica cells, cork cells and unclear identity cells) in all the cells in the silica–cork cell file. The cer-s.31 mutant was most severely clustered, with a 32% increased cell index compared with Bonus ($P < 0.05$; Fig. 2j) and clusters containing more cells (Fig. 2k), compared to cer-g.10, which had a 12% increased cell index ($P < 0.05$) and fewer cells in each cluster (Fig. 2j, k). We observed similar patterning defects on the lemma (Supplementary Fig. 8h) and similar but less severe trends in the silica–cork cell files in non-costal regions of the abaxial leaf sheaths (Supplementary Fig. 8i, j). Lastly, ectopic stomata also sometimes developed in the mutant silica–cork cell files (Supplementary Fig. 8g), suggesting that HvYDA1 and HvBRX-Solo may promote silica mother cell identity by suppressing stomatal identity. To confirm that all epidermal patterning defects occurred together in cer-g and cer-s alleles, we phenotyped 11 cer-g and 8 cer-s allelic mutants having premature stop codon, amino acid replacement and frame-shift mutations. All alleles showed equivalent stomata, prickle hair and silica–cork cell phenotypes (Supplementary Data 2; Supplementary Fig. 7b). Collectively our evidence suggests that

HvYDA1 and HvBRX-Solo act downstream of the epidermal lineage determination process to reinforce ACD-derived one-cell spacing for multiple cell types, determinacy and silica mother cell differentiation as well as influencing the plasticity of stomatal development in response to environmental cues.

## *HvYDA1* and *HvBRX-Solo* gene functions interact

Given that cer-g and cer-s mutants share a suite of distinct phenotypes, we explored genetic interactions. We crossed the cer-g.10 and cer-s.31 mutants, and then crossed resulting cer F$_2$ progeny with cer-g.10 and cer-s.31. As described in Supplementary Note S1, descendent progenies from the latter crosses contained the double mutant. In the field grown F$_2$, the wax phenotype of the single-mutant parents and the double mutant could not be visually distinguished from one another (Fig. 1a; Supplementary Fig. 1a, b) resulting in a modified 9:7 (wild-type:cer) ratio inferring that Cer-g and Cer-s act in the same pathway and are genetically linked (13.5 cM; 67 Mbp; Please see the Supplementary Note S1 in the supplementary information). We further examined epidermal and cuticular phenotypes in various combinations of mutant alleles (Fig. 3a–d; Supplementary Data 6) after genotyping of each locus (Supplementary Fig. 10). Double mutants showed wax phenotypes similar to cer-s.31, and stomatal, prickle hair cell and silica cell patterning defects in cer-g.10 cer-s.31 double mutants often resembled either parent in terms of cell index and more severe clustering, suggesting possible epistatic to additive/synergistic interactions (Fig. 3a–d; Supplementary Data 6). Strikingly, however, a single copy of the cer-s.31 mutant allele modified cer-g.10/cer-g.10 homozygous plants such that cer-g.10/cer-g.10 cer-s.31/+ individuals were either equivalent to cer-s.31 individuals or intermediate between cer-s.31 and cer-g.10 cer-s.31 double mutants (Fig. 3a–d; Supplementary Data 6). Heterozygous individuals of both genotypes were wild type in accord with the earlier classification of these mutants as recessive[37,40]. Taken together, genetic and phenotypic analyses suggest that these two genes function in a common pathway in controlling epidermal features.

## HvYDA1 and HvBRX-Solo regulate a cuticle and epidermal patterning enriched transcriptome

To learn more about the downstream pathways controlled by HvYDA1 and HvBRX-Solo, and how these relate to each other, we performed RNA sequencing (RNA-seq) of the basal portion of the second leaves of BW111, BW122 and Bowman (Fig. 4), a tissue with differentiating epidermal cells and which expresses *HvYDA1* and *HvBRX-Solo* (Supplementary Fig. 6). We used the BWNIL lines since we have extensive experience mapping transcriptomes in the Bowman background. We identified 685 HC differentially expressed genes (DEGs) in BW111 and 1,882 HC DEGs in BW122 compared with Bowman (Supplementary Data 7), consistent with the more severe phenotype of the cer-s.31 compared to cer-g.10 mutant. BW111 shared 72% of its DEGs (491/ 685) with BW122, where 403 are co-down-regulated and 81 are co-up-regulated, supporting a functional overlap between HvYDA1 and HvBRX-Solo (Fig. 4a). In agreement with their cuticular integrity phenotypes, a Gene Ontology (GO) enrichment analysis of the co-down-regulated DEGs showed a significant enrichment for terms associated with cuticle development; wax, cutin and suberin biosynthesis; and fatty acid transport (Fig. 4b; Supplementary Data 8), reflecting genes such as the β-ketoacyl-CoA synthases encoded by *HvKCS1*[49] and *HvKCS6*[50] and orthologues of *CER1* and *CER3* genes which encode enzymes that catalyse alkane formation[51,52] (Fig. 4c, e; Supplementary Data 8, 9). Notably, a *HvMYB96* transcription factor transcript whose Arabidopsis orthologue directly activates multiple wax biosynthesis and transport genes[53], was down-regulated in both mutants. We also detected expression changes in the orthologues of epidermal differentiation and patterning regulators, including known components of the YDA-signalling pathway, upregulation of transcripts encoding

a

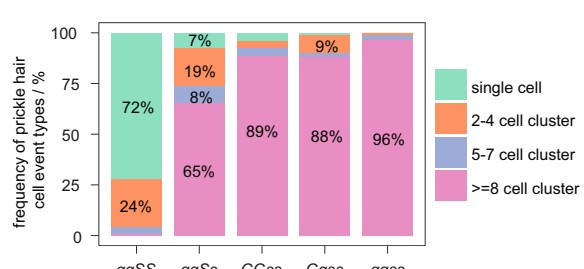

| genotype | *ggSS* | *ggSs* | *GGss* | *Ggss* | *ggss* |
|---|---|---|---|---|---|
| lemma wax | *cer-g* like | *cer-s* like | *cer-s* like | *cer-s* like | *cer-s* like |
| leaf stomatal clustering [1] | 5.1 % [b] | 9.2 % [a] | 1.9 % [c] | 3.8 % [bc] | 12.8 % [a] |
| leaf prickle hair cell index [2] | 60.4 % [c] | 94.5 % [b] | 96.8 % [ab] | 96.7 % [ab] | 99.5 % [a] |
| sheath silica/cork cell index [3] | 74.2 % [d] | 81.6 % [c] | 83.9 % [bc] | 85.9 % [ab] | 86.7 % [a] |

b

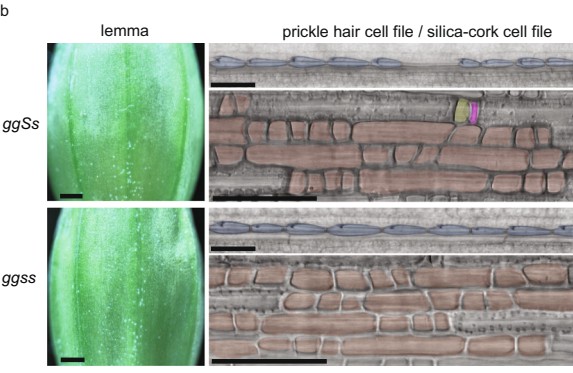

c

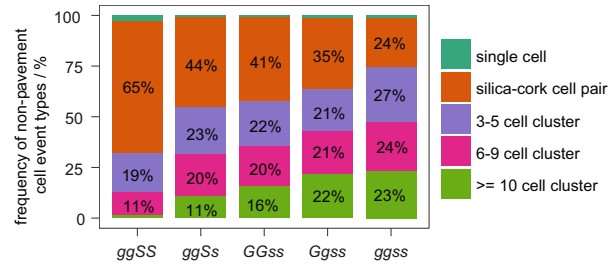

**Fig. 3 | Phenotypic analysis of *cer-g.10* and *cer-s.31* allele combinations.**
**a** Phenotypic severity of *cer-g.10/ cer-g.10* +/+, *cer-g.10/ cer-g.10 cer-s.31/*+, +/+ *cer-s.31/ cer-s.31*, *cer-g.10/ cer-g.10 cer-s*.31/+ and *cer-g.10/ cer-g.10 cer-s.31/cer-s.31*. Different letters indicate a significant difference ($p = 0.05$; [1]one-way ANOVAs with square-root transformation and Tukey's HSD multiple comparison; [2]Dunn's test following a Kruskal-Wallis test; [3]one-way ANOVAs and Tukey's HSD multiple comparison). **b** Lemma wax deposition (scale bars = 1 mm) and prickle hair cell (blue; scale bars = 100 μm) and silica (pink) -cork (yellow) cell patterning (scale bars = 50 μm) of *cer-g.10/ cer-g.10 cer-s*.31/+ and *cer-g.10/ cer-g.10 cer-s.31/cer-s.31*. Non-pavement cells with less defined silica–cork identity are coloured orange. $N = 8$

plants per genotype. **c** Stacked bar graphs numbers show frequency of single and clustered (two to four, five to seven and greater than eight) prickle cell hair events in *cer-g.10/ cer-g.10* +/+, *cer-g.10/ cer-g.10 cer-s.31/*+, +/+ *cer-s.31/ cer-s.31*, *cer-g.10/ cer-g.10 cer-s*.31/+ and *cer-g.10/ cer-g.10 cer-s.31/cer-s.31*. $N = 8$ plants per genotype.
**d** Stacked bar graphs showing frequency of normal silica–cork cell pair and abnormal cell patterning (single cell, cluster of three to five, six to nine and greater than ten non-pavement cells) events in costal (over vasculature) files of *cer-g.10/ cer-g.10* +/+, *cer-g.10/ cer-g.10 cer-s.31/*+, +/+ *cer-s.31/ cer-s.31*, *cer-g.10/ cer-g.10 cer-s*.31/+ and *cer-g.10/ cer-g.10 cer-s.31/cer-s.31* leaf sheaths. $N = 8$ plants per genotype. Source data, including $p$ values of statistic tests of **a**, **c** and **d** are provided in the Source data file.

orthologues of MUTE and ICE1, consistent with increased guard cell divisions and abnormal and supernumerary subsidiary cells, and changes in genes whose orthologues promote epidermal hair identity in maize[11], including several genes encoding SQUAMOSA-PROMOTER BINDING LIKE PROTENS (SPLs) and those involved in auxin biosynthesis and transport, agreeing with increased prickle hair cells (Fig. 4d; Supplementary Data 8). The co-down-regulated DEGs were also enriched for GO terms describing the responses to abiotic and biotic stimuli, including light (Fig. 4b, Supplementary Data 8, 9). We validated six DEGs by RTqPCR (Fig. 4e).

Altogether, the loss of HvYDA1 and HvBRX-Solo functions caused broad transcriptional reprogramming in suites of genes related to: (i) cuticle development, wax and cutin metabolism and transport, (ii) stomatal fate entry, (iii) the recruitment of SCs and GC terminal divisions, as well as (iv) factors involved in hair cell development, providing molecular insight into how they regulate multiple features during epidermal patterning and differentiation.

## Variation of *HvYDA1* and *HvBRX-Solo* in wild and cultivated germplasm
Grasses show substantial variation in glaucousness[28,54]. We explored *HvYDA1* sequence variation in 440 whole-exome sequences and a pan-genome consisting of diverse wild (*H. spontaneum*), cultivated (*H. vulgare*) and landrace barley accessions[43,55,56]. This revealed variation in 35 *HvYDA1* exonic sites (27 within coding regions; Supplementary Data 10), including nine non-synonymous variants outside the predicted kinase domain (Fig. 5a), of which two altered serine or threonine residues that represent potential regulatory Glycogen Synthase Kinase-3 (GSK3) phosphorylation target sites, shown to possibly

regulate YDA in Arabidopsis[57,58]. The variants were arranged into 30 haplotypes (HPs) (Fig. 5b; Supplementary Data 10) that separated into two subsets, A and B (Fig. 5b). Notably, subset A includes all HPs carrying gain-of-phosphorylation-site variants (Fig. 5a, b). Each of the four major haplotypes (HP_3, HP_10, HP_13 and HP_17) comprises over 45 accessions, with HP_3, HP_10 and HP_17 representing all genotype groups (*H. spontaneum*, two-rowed cultivars, six-rowed cultivars, two-rowed landraces and six-rowed landraces). HP_13 does not contain any *H. spontaneum* accessions, potentially indicating that HP_13 emerged after domestication and was retained during cultivation but could also reflect that a limited wild sample size. HP_25 appears to be derived from HP_13 but contains three unique non-synonymous changes (Fig. 5a), including T191S and P255L in the N-terminal region, and T747L in the C-terminal domain. Both T191S and T747L change the second Thr/Ser in Ser/Thr-x-x-x-Ser/Thr motifs, both of which are putative glycogen synthase kinase (GSK) target sites, but only the T191 residue is highly conserved across the YDA clade. Overall genetic similarity of *HvYDA1* reduced with an increasing geographical distance of accessions ($P < 0.01$; Supplementary Fig. 11), consistent with the geographic structuring of landrace HPs, where subset A HPs, including those carrying putatively extra phosphorylation target sites, tend to settle in the Middle East, while subset B HPs concentrate around the Mediterranean Sea (Fig. 5c). Intriguingly, we observed that of the nine accessions carrying HP_25, one originated in the Arabian Desert and five came from arid or drought-prone climatic zones of Ethiopia (Fig. 5c; Supplementary Data 11).

The *HvBRX-Solo* locus was not included in the Morex V1 assembly used to design the whole-exome sequencing capture; however, we

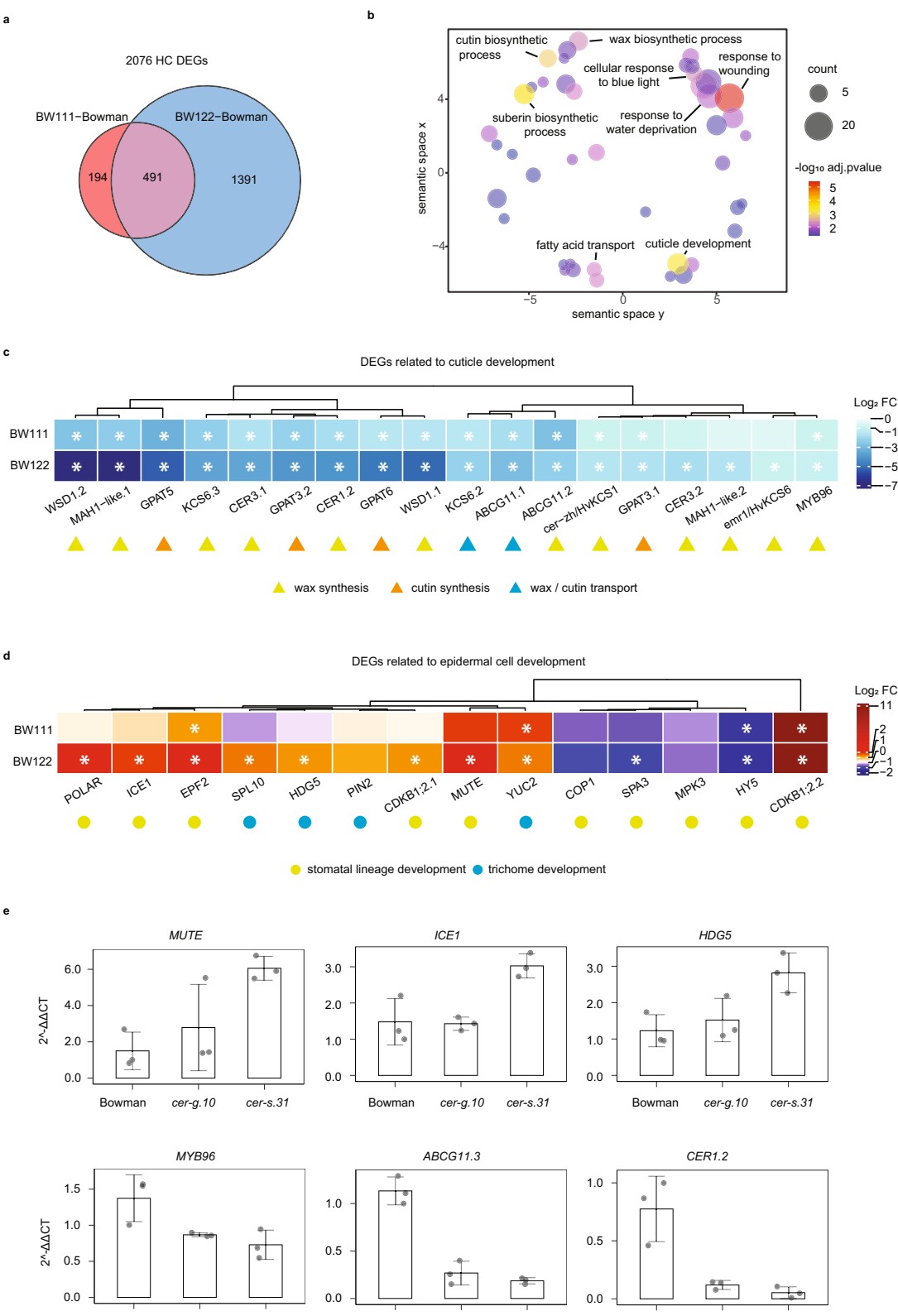

retrieved *HvBRX-Solo* genomic sequences from the 20 genomes assembled in the current pan-genome using the same pipeline as Morex V2[43]. Two lines showed variation in seven SNPs to form three haplotypes (Supplementary Data 12). One SNP is synonymous and occurs in the BRX domain, while five other SNPs cause alternative amino acids outside the BRX domain. Another variant contains a 6-bp in-frame deletion removing two amino acids, again outside of the BRX

domain. The functional significance of these changes is unclear. Collectively, while the conserved known enzymatic domains of *HvYDA1* and *HvBRX-Solo* in diverse *H. spontaneum*, *H. vulgare* cultivar and landrace genotypes showed little evidence of severe functional differences between HPs, we detected non-synonymous variants in putative regulatory residues as well as high geographic structuring in *HvYDA1*.

**Fig. 4 | Variation in *HvYDA1* and *HvBRX-Solo* causes similar expression changes in genes involved in epidermal development. a–d** Analyses of the RNA-seq of basal leaf sections from BW111, BW122 and Bowman. **a** Venn diagram of the overlap of co-differentially expressed genes (co-DEGs) between BW111 versus Bowman and BW122 versus Bowman. **b** Semantic similarities between significantly enriched gene ontology (GO) categories in downregulated co-DEGs in BW111 and BW122 versus Bowman. Bubbles represent GO terms whose closeness corresponds with semantic similarity. Bubble colours represent p values and bubble sizes represent the number of DEGs carrying this term. Selected representative terms of each cluster are labelled. **c**, **d** Heatmaps based on expression fold changes of representative DEGs related to cuticular development (**c**) and epidermal cell development (**d**) in BW111

and BW122 versus Bowman (Full list of DEGs is shown in Supplementary Data 7). DEGs with expression fold change ≥ 2 and a Benjamini−Hochberg adjusted $p$-value ≤ 0.05 are highlighted by asterisks. Genes encoding barley orthologs of MAPK3, SCAR, PIN2 and COP1 showed fold changes between 1.5 and 2.0 (Supplementary Data 7, 8) and were also taken as DEGs. **e** qPCR validation of selected DEGs: *MUTE*, *HORVU2Hr1G020990*; *ICE1*, *HORVU4Hr1G008480*; *HDG5*, *HORVU1Hr1G050620*; *MYB96*, *HORVU4Hr1G023510*; *ABCG11.3*, *HORVU2Hr1G090960*; *CER1*, *HORVU1Hr1G039820*. Expression data are expressed by $2^{-\Delta\Delta CT}$. Bars, error bars and points indicate means, standard deviations, and individual replicate values, respectively. $N = 3$ bio-replicates per genotype. Source data including $p$ values of statistic tests of **c**−**e** are provided in the Source data file.

## YDA and BRXf proteins contain land plant–specific motifs

HvYDA1 and HvBRX-Solo control epidermal features crucial for plant terrestrialisation. Searching for predicted *YDA* genes in the genomes of green algae (chlorophytes or charophytes), the closest relatives of the land plants, and different land plants (angiosperm, gymnosperm, fern, lycophyte and bryophyte species) showed that the YDA-like Ste11 class *MAPKKK* genes are likely specific to land plants. We constructed a phylogenetic tree of predicted protein sequences, using the top-hit charophyte sequences as the outgroup (Supplementary Fig. 12). Consistent with a published Arabidopsis MAPKKK phylogeny[59], AtYDA phylogenetically adjoins AtMAPKKK3 and AtMAPKKK5 and is distant from MAPK/ERK KINASE KINASE1 (AtMEKK1) and ARABIDOPSIS NUCLEUS- AND PHRAGMOPLAST-LOCALIZED KINASE1-RELATED PROTEIN KINASE1 (AtANP1). Comparing YDA and MAPKKK3/5 proteins revealed a high conservation of the kinase domain and defined 16 YDA-specific motifs outside the kinase domain, which were differentially distributed across land plant groups but not found in the charophytes, except for one motif in a *Mesotaenium endlicherianum* accession (Fig. 6; Supplementary Fig. 12). The YDA proteins in the gymnosperm and angiosperm species contained between 10 and 16 YDA-specific motifs, while the bryophytes contained six and lycophyte and fern YDAs had four and three motifs, respectively. The angiosperm and gymnosperm YDAs contained four highly conserved motifs in the N-terminal auto-inhibition regulatory domain of AtYDA, a region that also interacts with AtBASL[17]. The bryophyte and fern YDAs shared two of these four motifs with the lycophyte *Selaginella* YDAs. Based on this distribution, we suggest that YDA-specific motifs differentiate YDAs from other MAPKKKs in land plants.

In agreement with Koh et al.[60], we did not detect genes encoding BRXf proteins in either the chlorophyte or charophyte green algae, but found these genes in all land plant groups, with most species containing a pair of duplicated genes encoding BRXf proteins (Supplementary Fig. 13; Supplementary Data 13). Since the BRXf proteins are disordered outside of the BRX domains, we built a phylogenetic tree with the N-BRX domain sequences from the BRXf proteins selected from representative species, using a *Chara braunii* PRAF-like protein containing a BRX domain as the outgroup (Supplementary Fig. 13). Most land plants share the same BRXf protein structure with distinctive N-terminal motifs, including a s-acylation site within one of the most highly conserved motifs across all land plants in the tree, and two tandem BRX domains (Fig. 6; Supplementary Fig. 13). All grasses also contain a gene encoding a protein with a single BRX domain, such as *HvBRX-Solo*, characterised by the N-terminal BRX motif and a single BRX domain phylogenetically closer to the N-BRX domain of other BRXf members. We also detected single BRX domain encoding genes in: water ferns, *Arabidopsis thaliana*, where this gene (AT2G21030) is only expressed in the ovule, and in the gymnosperm *Pseudotsuga menziesii*, whose gene encodes a single BRX domain phylogenetically closer to the C-BRX domain of other BRXf members, and divergent from other land plants. Taken together, our phylogenetic analyses suggests that genes encoding YDA and BRXf proteins may have originated in the most recent common ancestor of land plants, aligning with the ancient origin and monophyly of stomata[61] and show

expansion of divergent motif structures in different plant groups (Fig. 6; Supplementary Figs. 12 and 13).

## Discussion

Our work shows that HvYDA1 and HvBRX-Solo control multiple surface characteristics in barley, including one-cell spacing between specialised cell types in the barley epidermis (Fig. 2). Rather than controlling each lineage's ACD and specialised cell fate per se, HvYDA and HvBRX-Solo may reinforce asymmetric cell fate following precursor ACD (Fig. 7), similar to the suggested role for BdYDA1[24]. However, rare appearance of ectopic stomata within silica cell files and missing silica−cork cell pairs in *cer-g* and *cer-s* suggests that HvYDA1 and HvBRX-Solo promote silica mother cell fate, division and daughter cell differentiation (Fig. 2; Supplementary Fig. 8). Profiling cell−specific and HvYDA1- and HvBRX-Solo-dependent changes in silica−cork, stomata and prickle hair cells may help resolve factors necessary to establish and reinforce distinct cell file lineages and their differentiation.

Both barley and Brachypodium have two *YDA* genes. *HvYDA1* is the closest orthologue of *BdYDA1* (BRADI_5g18180v3) while *HvYDA2* is the closest orthologue of *BdYDA2* (BRADI_3g51380v3). Defective alleles of *HvYDA1* caused relatively mild height and spacing phenotypes in *cer-g* mutants compared to the severely dwarf and highly clustered *Bdyda1* mutant[24]. Since *BdYDA2* is expressed at ~20% of the *BdYDA1* level in leaves (E-MTAB-4401 Array Express[62]) and showed no phenotype when knocked-out[24], and *HvYDA2* is expressed at relatively higher levels compared to *HvYDA1* (Supplementary Fig. 6d), we speculate that the milder phenotypes of *cer-g* could reflect a better ability of HvYDA2 to compensate for impaired HvYDA1 function. Isolating *Hvyda2* mutants and double mutant analyses will help evaluate this possibility.

YDAs regulate stomatal spacing in both dicots and grasses, yet the underlying mechanisms may be different. BASLs are specific to eudicots and were shown in Arabidopsis, and more recently in tomato, to direct YDA towards the larger daughter cell, likely through tethering to BRXf proteins[17,25]. Grasses do not contain *BASL* genes but along with other angiosperms, have *BASL-like* genes (*BSLLs*) that encode proteins sharing BASLs' D3 domain containing PxPF residues important for MAP kinase docking[63]. However, BSLLs are not known to associate with YDA, do not have the conserved D2 domain important for BASL polarity and do not show polar localisation when expressed in *Nicotiana benthamiana*[21,25]. Thus, there is little evidence that BSLLs could function akin to BASLs in YDA localisation. However, while barley lacks *BASL* genes, our data show that a single BRX domain containing protein, HvBRX-Solo, functions in a grass to ensure cell spacing in the epidermis. We are particularly interested in the molecular mechanism of single BRX domain proteins. Koh et al.[60] proposed that the tandem BRX domain arrangement is essential for robust protein localisation; however, AtBRX variants with deletion of either the N-BRX or C-BRX domain retain partial abilities to polarise its own and BASL's membrane localisation, and rescue the *brx* null mutant[21]. The single BRX domain of the PRAF factors were also recently shown to interact and polarise BASL[22]. BRXf proteins possibly anchor to the peripheral membrane via s-acylation[21], using a residue conserved in HvBRX-Solo.

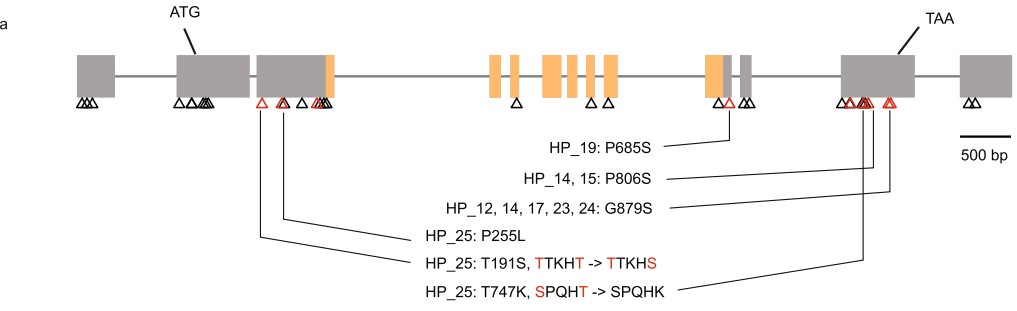

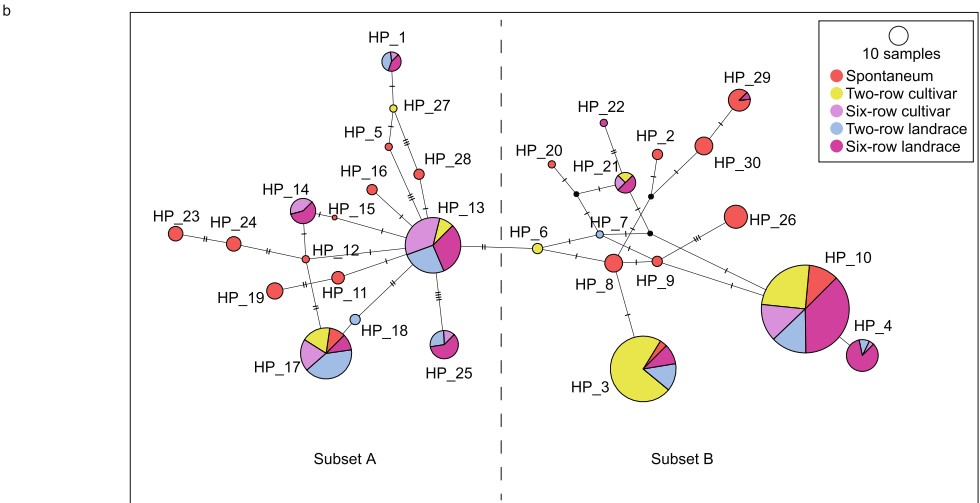

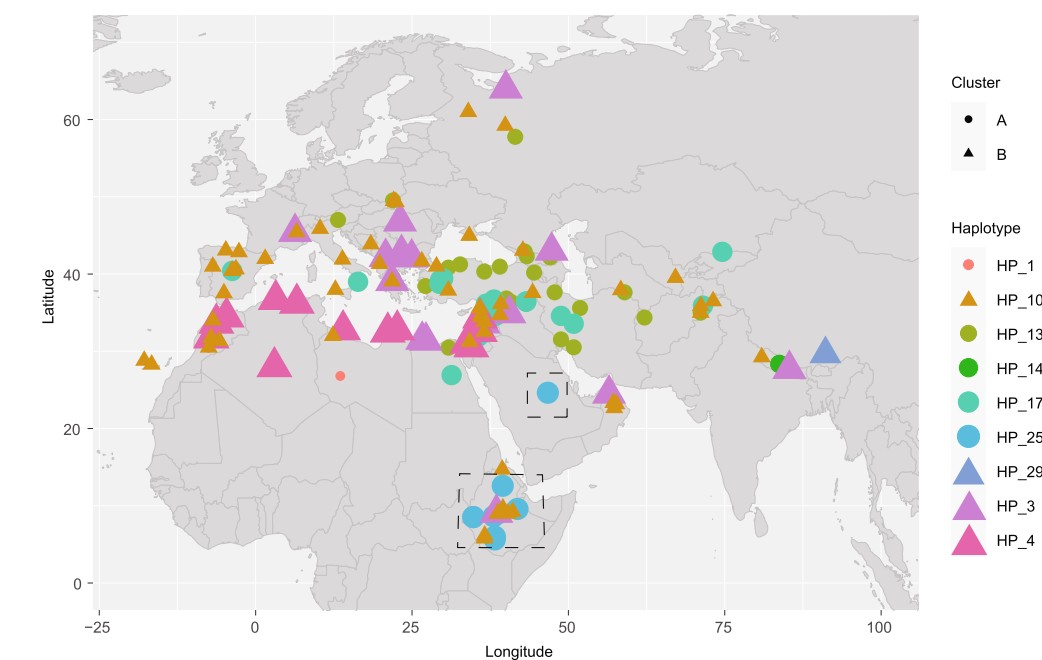

Given the role of s-acylation in responding to environmental and peptide signalling pathways[64], we are intrigued whether s-acylation may regulate HvBRX-Solo localisation and/or modify epidermal development in response to cues. Whether HvBRX-Solo directly interacts with YDA or another intermediary protein other than BASL to direct YDA function or otherwise promote pavement cell identity in the larger daughter cell is completely unknown. Unlike the single copy

of YDA in Arabidopsis, both AtBRXf and PRAF-encoding genes are part of multi-gene families which show limited levels of genetic redundancy in controlling stomatal spacing and other traits[21,22,65]. Multiple genes encoding tandem BRXf proteins are also present in barley. Single and higher-order mutant analyses of these families may reveal roles in regulating asymmetric cell fate in grasses. Determining (i) the interaction partners of HvYDA1 and HvBRX-Solo, (ii) whether HvBRX-Solo

**Fig. 5 | Geographic structuring of *HvYDA1* haplotypes. a** SNP sites within *HvYDA1* identified from the whole-exome sequencing data of 440 diverse *Hordeum spontaneum* (wild barley), *H. vulgare* cultivar and landrace accessions. Bars indicate exons while orange bars represent the kinase domain. Black triangles indicate noncoding region SNP sites and synonymous SNP sites while red triangles indicate nonsynonymous SNP sites. Haplotypes (HPs) carrying variants causing a loss or gain of potential phosphorylation sites as well as the extra nonsynonymous SNP site carried by HP_25, are noted below. Red letters indicate putative phosphorylation sites. **b** Median-joining network for *HvYDA1* HPs. The HPs are divided into subset A

and B, according to haplotypic relatedness. The node sizes are relative to the HP frequency. Bars between two nodes indicate the number of nucleotides within the sequence that differ between HPs. **c** Landrace HP distributions according to geographical origin. The underlying map was drawn using the R package "rworldmap". HPs are distinguished by colours and sizes. HPs from subsets A and B in **b** are represented by dots and triangles, respectively. Accessions carrying HP_25 are framed by dashed lines, while their geographic and climatic information are listed in Supplementary Data 11.

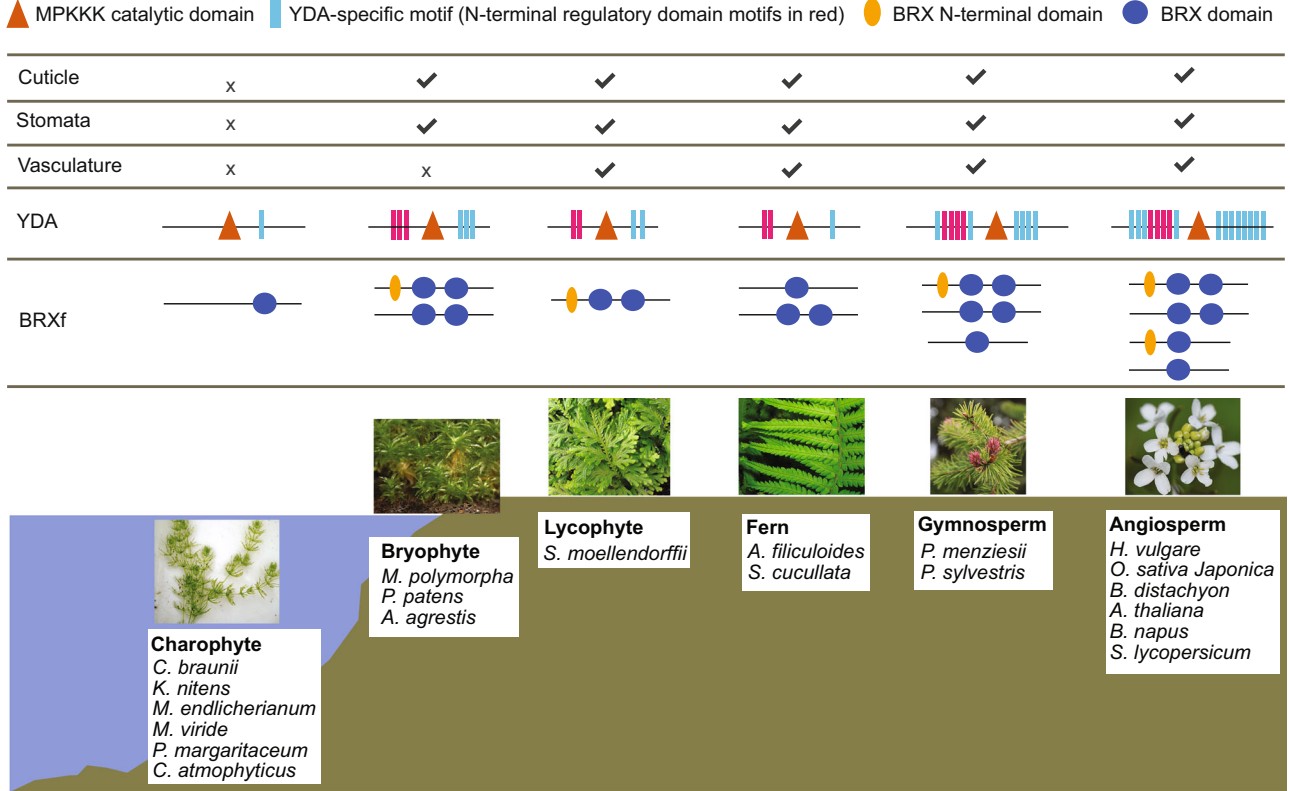

**Fig. 6 | YDA motif and BRX domain containing proteins appear together in land plants.** "X" and "✓" indicate the absence and presence, respectively, of each feature in different plant classes. Protein models of YDA motif and BRX-domain containing proteins summarised based on compositions of representative species in Supplementary Figs. 12 and 13. Representative pictures and the scientific names of species used for each class are indicated in boxes. Pictures of *Chara braunii*, and *Dryopteris*

*filix-mas* are attributed to Show_ryu and Rosser, respectively, and are covered under the CC-BY SA 3.0 license (https://creativecommons.org/licenses/by-sa/3.0/deed.en). Picture of *Arabidopsis thaliana* is attributed to Dawid Skalec and covered under a CC-BY SA 4.0 license (https://creativecommons.org/licenses/by-sa/4.0/deed.en). Picture sources are detailed in Supplementary Data 14.

and HvYDA1 colocalise or influence each other's distribution, as well as (iii) assessing differences between HvBRX-Solo and tandem domain BRX proteins in barley will help resolve the mechanism used by BRX-Solo proteins and their relationship to YDA in grasses.

Our work shows that HvYDA1 and HvBRX-Solo also promote cuticular integrity. Accumulating evidence from studies in Arabidopsis and other plants suggest links between cuticular properties and epidermal cell fate. For instance, the bHLH transcription factor ZHOUPI, interacting with ICE1, participates upstream of a MPK6-based cascade activated by a proteolytic processing of an embryo-derived peptide essential for continuity and integrity of embryonic and seedling cuticles[66–71]. Similar to *cer-g* and *cer-s*, mutants in this pathway show discontinuous 'patchy' cuticles and clustered stomata, suggesting that establishing a continuous cuticle may be important for stomatal spacing pathways to work properly[68,72]. On the other hand, recent work demonstrated that overly strong cuticles can also lead to stomatal clustering: overexpression of a SPCH target gene, *AtMYB16*, strengthened and thickened the cuticle and caused stomatal clustering, phenotypes linked to impaired AtBRXL2 polarisation and elevated cutin

biosynthetic gene expression, while transgenic cutinase expression rescued both the cuticular and stomatal phenotypes, pointing to possible interdependency between cuticle thickness, epidermal ACD and daughter cell fate[73]. These data align with a role in mechanical tension for AtBRXL2 responsiveness[74] and roles for MYB16 and SHN/WIN1 in epidermal cell specification and maturation in Arabidopsis and other species[34,73,75]. Thus, cuticular integrity and strength may be precisely tuned to ensure the effective function of pathways important to control cell reinforcement and differentiation (Fig. 7). Further, cuticular deposition may be regulated in tandem with cell fate specification, so that the distinctive cuticular chemistries and structures of pavement cells, trichomes/ hairs, stomata and silica cells are conferred by and reinforce their cell identities (Fig. 7). In support of this idea, AtMYB16 is normally highly enriched in the larger daughter cells and down-regulated by SPCH in guard cell precursors[76], suggesting that transcriptional cell fate regulators may match emerging cuticle properties in step with cell specification, elongation and maturation. In this way, downregulation of cuticular metabolic gene expression in *cer-g* and *cer-s* (Fig. 4c), may reflect and contribute to faulty reinforcement

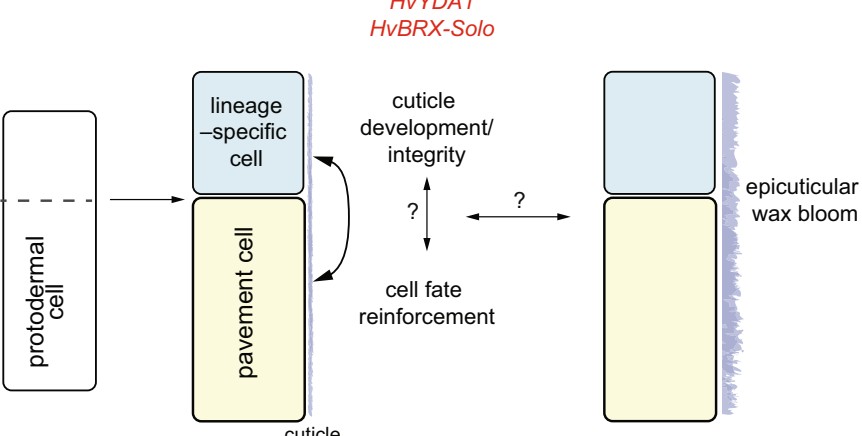

**Fig. 7 | Model HvYDA1 and HvBRX-Solo functions in barley epidermal features.** During epidermal development, protodermal cells undergo an asymmetric cell division to generate a specialised cell specific to the cell file identity (stomata, silica, prickle hair) and an epidermal pavement cell. HvYDA1 and HvBRX-Solo proteins reinforce asymmetric cell fate dictated by the lineage cues to prevent specialised fate in the epidermal pavement cell. HvYDA1 and HvBRX-Solo also control the cuticle properties in epidermal tissues. Both cell fate and cuticle development may interact with each other such that an incomplete or un-reinforced epidermal identity directly impacts the cuticular properties and vice versa. In this way HvYDA1 and HxBRX-Solo may mechanistically coordinate diverse epidermal features to ensure that emerging cuticle properties develop in step with cell specification, elongation and maturation. These activities may be important to explain the HvBRX-Solo and HvYDA1 control of DKS-derived aliphatic wax blooms in barley later in development.

of pavement cell fate in a feed-forward loop. Whether changes in cuticle development, composition and mechanical properties influence non-autonomous cell communication in *cer-g* or *cer-s* mutants is completely unknown; however, it would be interesting to learn whether the distribution, activity and/or perception of regulatory peptides, scaffolds or mobile factors such as MUTE are altered. Loss of function in a gene encoding a KCS enzyme underlies the Arabidopsis *hic* mutant, which does not show obvious cuticular wax defects but does increase its stomatal density in response to elevated $CO_2$, suggestive possible roles for lipid metabolism in epidermal fate. Moreover, beyond their cuticular roles, plant VLCFAs, including the abundant sphingolipids, are important structural membrane components and can act as intracellular signals and regulate intercellular communication[77], raising the possibility that precursors of the DKS pathway could also contribute to other metabolic and/or regulatory functions earlier in leaf development that influence patterning.

The *cer-s* and *cer-g* alleles were originally isolated due to their glossy phenotype. All of the *cer-s* and *cer-g* alleles we examined showed the same suite of epidermal patterning phenotypes, revealing that *cer-s* and *cer-g* alleles couple the glossy phenotype with the epidermal spacing defects. One explanation is that epicuticular defects observed in *cer-g* and *cer-s* reflect downstream or secondary effects from earlier defects on epidermal development (Fig. 7), especially as *HvYDA1* and *HvBRX-Solo* are expressed in young leaf sheath tissue, long before the DKS metabolic expression associated with the β-diketone and hydroxy-β-diketone deposition observed later in development (Supplementary Fig. 6). Cell type and cell–cell interfaces are proposed interact or potentially help orient epicuticular wax deposition on the epidermal surface, including a critical role for silica–cork cells in the extrusion of sorghum's wax bloom[78]. Notably, we observed silica–cork cell pairs and the wax bloom co-occur on reproductive stage tissues (leaf sheaths, elongated internodes and spikes) in barley. However, in maize the epicuticular wax and cell type specification typical of juvenile leaves are both coordinated by the GL15 locus in maize, but these activities were shown to be independent of each other based on sector analysis from GL15 activation in different tissue layers[79]. Inducible expression systems may help us understand whether defective epidermal fate and/or cuticular integrity may mechanistically impair epicuticular deposition later in development.

Regardless of whether cuticular integrity or epicuticular specialisations are primary or secondary effects of HvYDA1 and HvBRX-Solo control of epidermal patterning, multiple traits important for resiliency in terrestrial environments are controlled by single genes in barley. Land plants could benefit from coupling epidermal spacing of specialised cell types with cuticular development. Stomatal conductance directly connects transpiration-driven water transport to vapour exchange and carbon assimilation[5]. Increased vapour exchange from more efficient one-cell-spaced stomata may make cuticle impenetrability and the mitigation of water loss by epicuticular wax deposition even more important (Fig. 7). Some plants break the one-cell-spacing rule to produce clustered stomata, such as in certain *Begonia* and *Cinnamomum* species[80–82], while in some cases stomatal clustering can be induced by increased drought and salinity stress[83]. However, the molecular mechanism or adaptive value underlying plasticity or species variation in stomatal spacing is unclear. Given the role of MAP kinase cascades in environmental responses, a YDA-driven pathway could be involved in adjusting epidermal patterning in different conditions. We are also curious to assess potential functional consequences due to variation in putative GSK3 phosphorylation sites on HvYDA1.

Epidermal adaptations such as cuticles and stomata enabled colonisation of land by plants. Both specialisations control water loss, with stomata also enabling faster gas exchange in a plant body encased by the cuticle as well as efficient transpiration throughout the upward-growing plant body. Since xylem loading drives silica accumulation in the shoot, efficient transpiration may also underlie silica biomineralisation, a feature that protects land plants from biotic and abiotic stresses[84]. Epidermal hairs play multiple roles defending land plants from pests and low humidity, threats exacerbated by stomatal pores, also hinting at a functional relationship[85]. Our data show that YDA and BRX domain proteins regulate both epidermal patterning and cuticle formation and show distinct motif elaboration across land plants, consistent with the ancient monophyly of stomata and the origin and diversification of core cuticular and stomatal genes[61,86]. We speculate that these factors could participate in pathways which might fine-tune these adaptive features crucial for survival in the harsh terrestrial environment.

## Methods

### Germplasm, growth conditions and whole-plant phenotyping

Germplasm (Supplementary Data 2) of pertinent *cer* mutants and their wild-type cultivars were obtained from the Nordic Genetic Resource Center (Alnarp, Sweden). Plants were grown at either the University of Bristol (UK), the James Hutton Institute (UK) or at the University of Copenhagen (DK) with slight differences in growth conditions due to different equipment and space constraints. Alleles grown for DNA extraction, leaf sheath RNA isolation, toluidine blue staining and prickle hair and silica–cork cell phenotyping were grown in general purpose cereals compost under long day glasshouse conditions of 16h:8h light:dark cycle with day temperature of 18 °C and a night temperature of 14 °C, with supplemental light as required (James Hutton Institute, UK). For stomatal development phenotyping of *cer-g.10* and *cer-s.31*, the seedlings were grown in plugs containing a pre-soaked 3:1 mixture of sieved and autoclaved peat-based compost (William Sinclair Horticulture) and horticultural silver sand, placed in a Jumo Imago F3000 growth cabinet (Snijders Scientific) in 10h:14h light:dark cycle, set to provide a day photon irradiance of 120 μmol.m$^{-2}$.s$^{-1}$, 70% relative humidity, a day temperature of 22 °C and a night temperature of 20 °C (University of Bristol, UK). To phenotype the 3-d-old seedlings, sterilised seeds were germinated on damp filter paper in Petri dishes placed in growth cabinets under standard conditions. Phenotyping of architecture requires older or flowering plants, so 14-d-old plants were transferred to 18 cm pots and grown on in a temperature-controlled glasshouse (22 °C:20 °C day:night), with supplementary light when required (University of Bristol, UK). The internode number and height of the main stem were measured for 10 mature plants per genotype. The lengths of the five longest roots of 22 14-d-old plants were measured and averaged. For the $CO_2$ experiments, the plants were grown in growth cabinets in 12h:12h light:dark cycle, set to provide a day photon irradiance of 120 μmol m$^{-2}$ s$^{-1}$, 70% relative humidity and 20 °C constant temperature, at 400 ppm $CO_2$ or with elevated $CO_2$ (1000 ppm) (University of Bristol, UK). The plants used for leaf RNA-seq were grown in Snjider Microclima growth cabinets under 16h:8h light:dark cycle with day temperature of 20 °C and a night temperature of 16 °C, set to 200 μmol m$^{-2}$ s$^{-1}$ light irradiance, while plants used for the spike RNA isolation were grown under the same light conditions but at a constant 18 °C (James Hutton Institute, UK). The crossing between *cer-g.10* and *cer-s.31* was performed at University of Copenhagen (DK). The crossing progenies used for genetic interaction analysis, as well as 11 *cer-g* and 8 *cer-s* allelic lines (Supplementary Data 2) were grown in a speed breeding chamber (James Hutton Institute, UK) under 22h:2h light:dark cycle with day temperature of 22 °C and a night temperature of 17 °C.

### Epicuticular wax visualisation

The spikes, lemmas and leaf sheaths of plants at 18 days post anthesis (DPA) were examined under a Leica MZ FL3 dissection microscope at 8, 10 and 20× magnification, respectively. Fully developed leaves from 14-d-old plants, and spikes and flag leaf sheaths from flowering plants, were detached and air-dried for scanning electron microscopy. Small (5 mm long) sections were placed onto carbon conductive tabs on stubs, sputter-coated with platinum and viewed using a Quanta 400 scanning electron microscope (FEI Company) using 25 kV of voltage and a spot size of 4.0. Images were taken at 10,000× and 20,000× magnification.

### Toluidine blue staining

Several tissues from Bonus, *cer-g.10* and *cer-s.31* were detached and stained in 0.05% (w/v) toluidine blue. The spikes and leaf sheaths were harvested at 18 days post anthesis (DPA) and immersed in stain for 5 h or 24 h, respectively. Fully expanded second leaves were harvested 10 days post germination and immersed in stain and checked every 2 h for 6 h before a longer incubation of 28 h. At the

end of the immersion period, tissues were rinsed with water and photographed.

### Wax quantification

The first leaves of 14-d-old plants and spikes from mature plants were dipped twice into 99% HPLC molecular-grade chloroform for 15 s and then 10 s, an internal standard of 0.5 μg.mL$^{-1}$ tetracosane ($C_{24}$ *n*-alkane) was then added. Lemma awns were trimmed from the spikes prior to the analysis. Solvent was removed from the wax extracts under a gentle stream of $N_2$. Wax residues were derivatised using *N,O*-bis(-trimethylsilyl)trifluoroacetamide + 1% trimethylchlorosilane (BSTFA + 1% TMCS; Sigma-Aldrich) and excess reagent removed again under $N_2$. The wax constituents were identified by their electron-impact mass spectra (70 eV, *m/z* 50–950) after capillary GC (Restek Rxi-1HT, 15 m × 0.32 mm, 0.1 μm [Thames Restek Ltd.]) performed on an ISQ-LT GC–MS comprising a 1300 gas chromatograph combined with a single quadrupole mass analyser (Thermo-Fisher Scientific). The samples were injected onto the analytical column at 50 °C, ramping at 0.2 °C s$^{-1}$ to a maximum of 380 °C, where it was held for 7 min. The flow rate of the He carrier gas was 5 mL min$^{-1}$. All data were acquired using the XCalibur software suite (Thermo Fisher Scientific). Leaf area and spike length were calculated using ImageJ[87].

### Bulked segregant genotyping and exome capture

Equal amounts of DNA from cv. Bowman, cv. Bonus, BW111, BW122 and $F_2$ individuals showing mutant phenotypes (54 from BW111 × cv. Morex and 14 from BW122 × cv. Morex) were pooled for a total of 300 ng and genotyped using the Barley 50K iSelect SNP chip[41]. Polymorphic sites between Bowman and Bonus where the mutants genocopied Bonus determined the introgression regions in both mutants. For the bulked segregant analysis, polymorphisms between the markers of both mutant bulks and the parents were determined. In the mutant bulks, the regions outside the introgressed locus display a proportion of 0.5 for each of the parent alleles, and are called as heterozygous. In regions flanking the causal locus in mutant bulks, the proportion of mutant alleles will be greater than 0.5 but less than 1.0, and the regions are called as heterozygous or missing. Closer to the causal locus, the mutant allele will predominate in mutant bulks to approach 1.0, and is called as a homozygous mutant. The data were visualised using R[88].

The BW111 × cv. Morex $F_2$ mutant bulk as well as BW111 itself were also subjected to exome capture sequencing[89]. Sequencing read quality was assessed using FastQC[90], after which the reads (2 × 100 bp) were mapped against the Morex pseudomolecule 150831 assembly[91] with BWA version 0.7.17[92]. A manual review of the mapping data was performed using Tablet[93]. Single-sample SNP calling was performed using the GATK Best Practices Pipeline[94]. The called SNPs were filtered using bcftools (SAMtools) using a minimum base quality of 100 and a minimum read depth of 30 before being visualised. Allele frequencies in bulk were calculated as the number of reads supporting the mutant allele divided by the number of reads at every polymorphic SNP position and visualised along the physical map of barley[95] using R. A rolling-average method was used to average the allele frequencies within a sliding window size of 100 bp.

### Fine mapping

From a BW122 × Morex $F_2$ population, 253 wild-type and 60 mutant plants were selected for fine mapping. Six KASP primers sets (Supplementary Data 15) were designed using the Web-based Allele Specific Primer (WASP) tool[96], with two designed at each border of the introgression on chromosome 2H and additionally one designed on each side of the chromosome between the centromere and introgression border. In a second round, 34 plants (16 with wild-type and 18 with mutant phenotype) showing recombination within the refined region were mapped with another six primer sets (Supplementary Data 15). Each KASP primer set consisted of two allele-specific primers as well as

one common primer. The allele-specific primers were fluorescently labelled (FAM or VIC). The genotyping reactions were performed using a PCR-based KASP genotyping assay kit (LGC Biosearch Technologies) according to the manual.

## Whole-genome sequencing

Genomic DNA from BW122, Bowman and Bonus were subjected to whole-genome sequencing on the Illumina MiSeq platform. Raw reads were mapped to the Morex v2 assembly[42] with BWA[92]. Sambamba[97] was used to remove duplicate reads and reads with more than six mismatches. Mapping data were manually reviewed using Tablet[93]. The GATK toolkit[94] was used to re-align indels and recalibrate variant calls, producing a recalibrated BAM file. Small variant calling of BW122, Bowman and Bonus was performed using freebayes[98] and structural variant calling was performed using Manta SV caller[99]. SNPs from the small variant caller were used to define the introgressed regions by comparing BW122 against Bowman and Bonus. Only variants within the Morex v2 gene models[42] of the introgression regions and for which the mutant carries a homozygous mutant allele that differs from Bonus and Bowman were retained.

## Candidate gene analysis and genotyping

Overlapping regions covering the complete genomic region of *HvYDA1* and *HvBRX-Solo* were amplified using OneTaq 2X Master Mix with Standard Buffer and OneTaq Hot Start 2X Master Mix with GC Buffer (New England Biolabs), respectively. To genotype the *cer-g.10* X *cer-s.31* crossing progenies, primer sets were designed at *cer-g.10* and *cer-s.31* loci, respectively, for amplifying wild type and mutant alleles of each locus. PCR reactions were performed using GoTaq® Flexi DNA Polymerase and green reaction buffer. Primer sets and PCR programs are listed in Supplementary Data 15. The amplicons for sequencing were cleaned with Exosap (Applied Biosystems).

## Epidermal cell imaging and quantification

For the stomatal imaging, 1 cm epidermal peels were taken from the abaxial surface of the first leaf of 14-d-old plants by making a shallow, horizontal cut with a razor blade across the leaf tissue which was then removed, leaving the lower epidermal cell layer which was imaged under a DM-IRB inverted microscope at 200× magnification. A stoma was considered "normal" if it was not directly next to another stoma and had properly formed guard cells and subsidiary cells. Subsidiary cell abnormalities were only recorded if the stoma was not part of a stomatal cluster. For the stomatal quantification, leaf impressions of both the abaxial and adaxial surfaces of the first leaf of 14-d-old plants were acquired using dental resin. Dental resin was applied to a detached leaf and left to set. Following the removal of the leaf material, clear nail varnish was applied to the putty and left to dry overnight. Leaf impressions were also made of both surfaces of the first leaf of 3-d-old plants and the flag leaf sheath of flowering plants. Leaf impressions were imaged at 100× magnification. Five independent cell files of stomata were examined per plant, with 25 stomatal events examined per cell file (*n* = 24 plants/ genotype with three biological replicates per measurement). A stomatal event was defined as either a single stoma, a stomatal cluster, or two pairs of guard cells flanked by a single subsidiary cell on one side. For the stomatal density and index calculation, the number of stomata, stomatal events and epidermal cells were counted in four separate 0.25-mm² areas per leaf (*n* = 32 plants/ genotype with three biological replicates per measurement). The lengths of 20 guard cells per plant were measured in Volocity (Quorum Technologies).

For the imaging and quantification of the prickle hair cells, the second leaf of 14-d-old plants, which showed more consistent growth compared to first leaves, and flag leaf sheaths of flowering plants were collected, while for the silica–cork cell pairs, the lemmas of heading plants were harvested. Materials of Bonus, *cer-g.10* and *cer-s.31* were placed into 7:1 ethanol:acetic acid and incubated overnight to remove chlorophyll, followed by a 5-h water rehydration. Rehydrated leaves, sheaths and lemmas were stained in 0.05% toluidine blue for 5 min, 1 min and 30 s, respectively, then rinsed in water. The stained leaves were mounted on slides to image the costal regions in eight 1.17 mm × 0.93 mm (1.1 mm²) views under 100× magnification. The numbers of prickle hair cells, pavement cells and prickle cell events in the costal regions of each leaf were counted. The stained sheaths were mounted on slides with the abaxial side facing up and each imaged in eight 0.60 mm × 0.40 mm (0.24 mm²) views under 200× magnification that captured costal and non-costal regions. The numbers of silica cell events and pavement cells in the costal regions of each leaf sheath were counted. A silica cell event was defined as either a normal silica–cork cell pair, a single silica cell or any number of cells in a cluster that were not identified as pavement cells. To calculate the percentages of silica cell events and pavement cells in non-costal regions, 80 continuous cell events on each image were examined, resulting in the examination of 640 cell events per sheath. Stained lemmas were mounted on slides with the abaxial side facing up and imaged under 400× magnification. Nail varnish impressions on the second leaf adaxial side and flag leaf sheath abaxial side were made for materials of other 11 *cer-g* and 8 *cer-s* alleles (Supplementary Data 2) and *cer-g.10* X *cer-s.31* crossing progenies. They were imaged as described for the stained samples. *cer-g.10* X *cer-s.31* crossing progenies were quantified for prickle hair cell patterning and stomatal clustering percentage on leaves and costal silica–cork cell patterns on sheaths. Stained leaf and sheath samples were also made for *cer-g.10* X *cer-s.31* crossing progenies and imaged for representing cell patterning.

## Transcriptome analyses

The second leaf of ten 10-d-old plants of Bowman, BW111 and BW122 were dissected and the basal section (bottom 3 cm above the seed) were pooled as one replicate for RNA isolation. Four replicates were made for each genotype. RNA isolation was performed using TRI Reagent (Sigma-Aldrich) as described[100], followed with a clean-up using the RNeasy Mini Kit (Qiagen). RNA integrity was confirmed with a Bioanalyzer 2100 (Agilent Technologies). A strand-specific mRNA library was constructed and then sequenced on a NovaSeq 6000 PE150 platform at Novogene UK with 2 × 40 million 150-bp paired-end reads for each replicate. Raw data were mapped to the Barley Reference Transcript BaRTv1.0[95] for quantification using Samtools[101]. Differential expression analysis was performed using 3D RNA-seq App[102]. During data pre-processing, CPM cut-off was set to 1 while the sample number cut-off was 2 to filter out low expression genes. During differential expression analysis, the limma-loom weights pipeline was used. All other settings were selected as recommended. Unless specified, only genes with an expression fold change ≥ 2 and a Benjamini–Hochberg adjusted *p*-value ≤ 0.05 between the mutant and Bowman were taken as differentially expressed genes (DEGs). Numbers of DEGs were counted based on BaRTv1.0 IDs. A GO enrichment analysis of the DEGs was performed on g:Profiler[103] using a custom barley GO annotation computed by citing the GO annotations of the top-hits (BLASTp, identity > 35% and query coverage > 70%) from *Arabidopsis thaliana*. The enriched GO terms were summarised using ReviGo[104] before the visualisation.

## RT-qPCR

RTqPCR validations for the RNA-seq data were performed using three replicates of the RNA used for the RNA-seq. In addition, RNA was isolated from florets of 29 days post germination (dpg) and 35 dpg plants. Flag leaf sheaths for RNA isolation were collected when their auricles were 1-3 cm above the auricles of the second-to-flag leaf sheaths. Three continuous 2 cm length sections initiated from the bottom, one 4 cm section with 1 cm above and 3 cm below the second-to-flag leaf sheath auricle and one 2 cm section right below this section were sectioned

from each sheath. Sections from six plants were pooled as one bio-replicate. The synthesis of cDNA was performed using the SuperScript VILO cDNA synthesis kit (Invitrogen) and 2 μg total RNA. Roche Fas-tStart Universal Probe Master mix (Rox) coupled with the Universal Probe Library probes #126 and #156 (Roche) were used to detect transcripts of *HvYDA1* and *HvBRX-Solo* in leaf and spike, respectively, with *HvActin7* (HORVU5Hr1G039850.3; probe #129) and *HvPP2A* (HORVU5Hr1G109430; probe #7) used as the endogenous controls. The SYBR Green Power Up kit (Thermo Fisher Scientific) was used for other qPCR experiments where *HvActin7* was used as the endogenous control for sheath section samples while an extra endogenous control, *HvREF2* (HORVU7Hr1G096480), was used for other experiments. The primers for the qPCR are listed in Supplementary Data 15. Three or four bio-replicates and three technical replicates were made for the quantification of each target. The reactions were run on the Applied Biosystems StepOne system. The expression values were determined using the $2^{-\Delta CT}$ (mean ± standard deviation) method relative to the endogenous control(s) or the $2^{-\Delta\Delta CT}$ method to present the expression levels across the genotypes, as indicated in the figures.

### Haplotype analysis
The SNP data of *HvYDA1* from the exome-capture datasets from *H. spontaneum* and the *H. vulgare* cultivars and landraces[55,56] were filtered to retain sites with ≥98% of samples homozygous. The *HvYDA1* and *HvBRX-Solo* genomic sequences of 20 diverse *H. spontaneum*, and *H. vulgare* cultivar and landrace accessions making up the barley pangenome[43] were also obtained. The exome-capture and pan-genome *HvYDA1* SNP data were merged, and only the exonic sites were retained; accessions with missing data points or heterozygosity at these sites were excluded. The resulting dataset containing the exonic SNP sites of 440 accessions was used for the haplotype analysis of *HvYDA1*. The Median-Joining haplotype network construction was performed using *PopArt* (http://popart.otago.ac.nz). *HvBRX-Solo* SNP data of 20 accessions were used for the haplotype identification. 355 accessions with available geographic coordinates were used for the spatial auto-correlation analysis to assess the relationship between the inter-individual genetic identities of pairs of accessions and the geographical distances for the *H. spontaneum* and *H. vulgare* landrace groups. The analysis was based on Ritland's kinship coefficient as a measure of genetic identity over geographical distances, undertaken using SPAGeDi[105] as described by Russell et al. (2016) with the following modification: the pairwise geographical distances between accessions were calculated using the R package "geodesic". Finally, 159 landraces with geographic co-ordinates were plotted on a world map using the R package "rworldmap". Climatic zones of Ethiopia were determined according to potential productivity, species composition and distribution of the plant community[106].

### Phylogenetic analysis
*Arabidopsis thaliana, Brachypodium distachyon, Brassica napus, Chara braunii, Hordeum vulgare, Marchantia polymorpha, Oryza sativa japonica, Physcomitrium patens* and *Selaginella moellendorffii* protein sequence data were obtained from EnsemblPlants (https://plants.ensembl.org/). *Solanum lycopersicum* data were obtained from the Sol Genomic Network (https://solgenomics.net/), while *Anthoceros agrestis [Bonn]* data were collected from the hornworts database of the University of Zurich (https://www.hornworts.uzh.ch/en.html). The *Azolla filiculoides* and *Salvinia cucullata* data came from FernBase (https://www.fernbase.org/); the *Pinus sylvestris* and *Pseudotsuga menziesii* data came from PLAZA Gymnosperms (https://bioinformatics.psb.ugent.be/plaza/versions/gymno-plaza/); the *Chlorokybus atmophyticus, Klebsormidium nitens, Mesotaenium endlicherianum* and *Mesostigma viride* data came from the Algae database (https://phycocosm.jgi.doe.gov/phycocosm/home); and the *Penium margaritaceum* data were downloaded from the Penium Genome

Database (http://bioinfo.bti.cornell.edu/cgi-bin/Penium/home.cgi;[107]). These data were used to build a local BLASTp database. Representative photos of the plant groups were obtained from the Wikimedia Commons, and the photo sources were described in Supplementary Data 14. HvYDA1 and HvBRX-Solo proteins were used as BLASTp queries against the database, with E-values of 1e-110 and 1e-5 used as the cut-offs for the YDA orthologues and BRXL orthologues, respectively. *Selaginella moellendorffii* ortholog sequences presenting potential assembly errors were manually corrected by aligning the genomic sequence to other YDA orthologues. The phylogenetic trees were constructed using MEGA X[108] based on the YDA catalytic domain and the BRX domain of the YDA and BRXf proteins, respectively, with the sequence alignment performed using ClustalW and the trees inferred using the neighbour-joining method. For the phylogeny test, 1000 bootstrap replicates were used, and the complete deletion option was selected to treat gaps and/or missing data. Conserved motifs among the proteins were explored using the MEME motif search tool (http://meme-suite.org/tools/meme). The motif search was performed with the default settings with the following modifications: maximum number of motifs: 50 for YDAs and 20 for BRXfs; optimum motif width: 6–28. The resulting motif structure of the proteins were aligned with the phylogenetic trees using TBtools, followed by manual edits.

### Data statistics and visualisation
All statistical analyses were carried out using R unless specified. Two-sided t-tests, Two-sided permutation test, one-way ANOVAs w/wo data transformation and Tukey's HSD multiple comparison or non-parametric Kruskal–Wallis tests coupled with post-hoc Dunn's tests were used, as specified in the legends. The data visualisation was conducted using R unless specified and assembled using Adobe Illustrator CC2017. Measurements were taken from individual biological replicates; no samples were measured repeatedly.

### Reporting summary
Further information on research design is available in the Nature Research Reporting Summary linked to this article.

## Data availability
All materials described in the paper will be made freely available. Source data are provided with this paper. RNA-Seq and exome capture data sets are deposited into the European Nucleotide Archive (ENA), http://www.ebi.ac.uk/ena, under this project accession number PRJEB53837 Source data are provided with this paper.

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

## Acknowledgements

We are greatly indebted to Dr. Sandy Hetherington (University of Edinburgh) for advice on phylogenetic analyses and land plant evolution, as well as to Drs Joanne Russell and Ian K Dawson for guidance on genetic variation analyses. S.M. and C.C. acknowledge funding from the Biological and Biotechnological Research Council (BBSRC, BB/R010315/1). M.S-D acknowledges support by the ERASMUS+ programme and his home university, Hogeschool van Arnhem en Nijmegen. L.L. was supported by the China Scholarship Council and the University of Dundee. T.M. was supported by a Carnegie-Cant-Morgan PhD Scholarship and the University of Dundee. S.J. acknowledges a CASE postgraduate studentship from the BBSRC with additional funding from James Hutton Limited. R.W., L.R., L.M. and M.B. were funded from the Scottish Government's Rural and Environment Science and Analytical Services Division Theme 2 Work Program 2.1. M.S. is funded by BBSRC ERA-CAPS BB/S004610/1. A.H. is grateful for support from the Leverhulme Trust (RPG-2019-004). The NERC is thanked for partial funding of the mass spectrometry facilities at Bristol (R8/H10/63). The authors acknowledge the Research/Scientific Computing teams at The James Hutton Institute and NIAB for providing computational resources and technical support for the "UK's Crop Diversity Bioinformatics HPC" (BBSRC grant BB/S019669/1), use of which has contributed to the results reported within this paper.

## Author contributions

Conceptualization, S.M., A.H. and C.C.; Methodology, S.M., A.H., C.C., M.B., L.R., R.W. and I.B.; Formal analysis, S.J., L.L., C.C., I.B., P.W-K.; Investigation: S.J., L.L., C.C., P.W-K., T.M, M.S-D., Y.Z. and M.E.; Data analysis: L.L., S.J., M.B., T.B., M.S. and L.M.; writing—original draft preparation, S.M. and L.L.; writing—review and editing, S.M., A.H., S.J., C.C., L.L., P.W-K., M.B, R.W. and T.M.; supervision, S.M., A.H. and C.C.; project administration, S.M. and A.H.; funding acquisition, A.H., S.M., R.W. and C.C. All authors have read and agreed to the published version of the manuscript.

## Competing interests

The authors declare no competing interests.
