## [Peer Review File · Nature Communications]

Conserved signalling components coordinate epidermal patterning and cuticle deposition in barleyREVIEWER COMMENTS

Reviewer #1 (Remarks to the Author):

What are the noteworthy results? Will the work be of significance to the field and related fields?

This paper reports the molecular identification of a classic Barley mutant *cer-g* whose phenotype of less wax and clustered stomata has featured among mysteries of plant development field for several decades. It presaged the link between the production of cuticular compounds and epidermal cell type identities, a link that has been strengthened recently by studies such as Yang et al., 2022, Plant Cell on cutin biosynthesis and stomatal identity. Among Barley *cer* mutants, *cer-g* was historically reported to be unique in its dual effect on stomata and wax production, but the authors have identified a second mutant, *cer-s* that shares some features with *cer-g*.

This work identified the causal gene for *cer-g*, and looks at wax and stomatal phenotypes and genetics from many angles. There are a number of things to really like about this paper. The introduction is well written and compelling, and the figures are well presented.

I have two main criticisms, one can be handled through writing, but the other requires additional experiments.

1) Major experimental concern. I am not entirely convinced that the *HvBRX* is the causal gene for *cer-s*, or at least that the developmental phenotypes are caused by the lack of the same gene as the wax phenotypes. The reasons for this are elucidated below, but I think this data really needs to be confirmed before anything else can be interpreted

2) Major writing concern. The ratio of data to discussion and speculation is tilted quite far in the direction of speculation. The speculation concerns the mode of action of the genes underlying *cer-g* and *cer-s*, the effect of mutations in *cer-g* on plant adaptation, and the role of these genes in evolution of the cuticle and life on land.

Some of the ideas reported in the results should really be in the discussion (and the speculation toned down considerably).

For example, lines 190-193, "Collectively our data reveal that *HvYDA1* and *HvBRXL1*, orthologues of genes which work together to control stomatal development in *Arabidopsis*, likely work together to promote cuticular integrity and wax deposition in barley". I disagree with the single BRX-domain protein being called an orthologue, that there is ANY evidence that BRXL genes and YDA work together as a complex in *Arabidopsis*, and am skeptical of the claim that the two work TOGETHER in barley.

Likewise, there is no actual evidence for "shared pathways" (line 253, line 275), nor are the clear data showing a "molecular insight into how *HvYDA1* and *HvBRXL1* regulate multiple features during epidermal patterning (line 301).

It is a somewhat smaller point, but the naming of the gene that contains a single BRX domain as *BRXL1* will cause confusion in the field. In part, because the term BRXL already is used for 2 BRX-domain proteins for at least a decade (Hardtke lab papers) and this nomenclature is continued in other labs. BRX domains are present in other proteins as well, such as the PRAFs (recently described in Wang et al., 2022, Nat Comm). The mechanisms by which PRAFs and BRX family proteins work are very different and without any experimental data showing functions or interactions of the Barley single BRX-domain with other proteins (such as YDA), the idea of a YDA/*HvBRX* complex needs to be toned down.

Sections about the adaptive significance of HvYDA alleles in natural populations should be moved out of the results section as there are no functional or pop-gene analyses of these alleles.

- Does the work support the conclusions and claims, or is additional evidence needed?

In general, the phenotypic analysis of wax composition and distribution and of epidermal features is well done on the genotypes in which it is done.

The mapping of these genes is a heroic task, but there is little validation of gene identity. I do not expect that there would be complementation or functional studies in Barley, but there needs to be some more convincing data that causal genes for cer-g and cer-s are indeed HvYDA and HvBRXL1, and that the stomatal and wax phenotypes are due to disruption of the same gene. As far as I can tell from Sup table 2, only two cer-g and one cer-s line have stomatal clustering defects. Many lines with mutations disrupting the loci were only measured for wax.

I read the description of mapping and the creation of the double cer-g cer-s. As much as I love the classical genetics used here, as far as I can tell the double mutant was never confirmed in terms of molecular defects in the HvYDA and HvBRXL1 loci.

In Figure 2, the phenotypes of the double cer-g cer-s mutant sometimes looked additive and sometimes the cer-g phenotype predominates. These results are hard to reconcile with a model of their gene products working as a complex to guide both epidermal fate and wax deposition. In fact, what they really look like to me is that the epidermal fate and proliferation phenotypes of cer-s are due to another mutation loosely linked, and in the crosses, they have lost this second mutation.

To me, without a solid foundation that cer-s is HvBRXL1 and this locus is responsible for the epidermal phenotypes, the rest of the work is hard to interpret. But I do think there are some fairly straightforward ways to fix this—A and B, below.

A. Measure phenotypes in trans-heterozygotes (F1 progeny of two cer-s lines), using mutants with nonsense or frameshift mutation in HvBRXL from among the list in Table S2. Whenever possible, the neighboring loci shown in Table S4 should be checked to make sure they don't contain mutations. If the epidermal development AND wax phenotypes are seen in the trans-hets, I would be confident that the correct gene was identified.

B. Not as good an experiment as A, but faster, would be to quantify the epidermal phenotypes in at least 3 of the with nonsense or frameshift mutation in HvBRXL from among the list in Table S2. Doing the same with an additional line of HvYDA with a nonsense mutation (rather than 3 bp change that affects two amino acids in the kinase domain) would make it somewhat clearer that the cer-g phenotype was due to loss, rather than reduction of HvYDA function.

Other data concerns:

Figure 3 is fine in terms of the data presentation, but the title: Variation in HvYDA1 and HvBRXL1 causes similar expression changes in genes involved in epidermal development does not seem to fit the data in e. where the magnitude of change genes in cer-g and cer-s is quite different. The gene expression appears to be a perfect match for the strength of the phenotype, and I don't have any concerns with the data itself. I do think it is quite weak evidence that the genes underlying cer-g and cer-s act in a common pathway. Two completely independent pathways that resulted in wax and stomatal phenotypes would give the same transcriptional results.

Figure 4. As a representation of allele distribution this is fine. The suggestion that mutations affecting phosphorylation sites have any meaning is dubious.

Figure 6. HvYDA1 and HvBRX are shown as if acting as a complex. But there is no evidence in any species that they would. Even taking the best characterized systems in Arabidopsis, BRX does not interact with MAPKKs, but rather with D6PKs, and BASL, a protein that is not outside of angiosperms, is the glue between these proteins. A complex of the two proteins working together exclusively is inconsistent with the divergent phenotypes in the cer-g and cer-s lines and reported epistatic effect of cer-g.

- Are there any flaws in the data analysis, interpretation and conclusions? - Do these prohibit publication or require revision?

Many of the interpretations go quite a bit beyond the data and, in particular with HvBRX, make a fair bit of assumption. The BRX genes were also quite well studied in roots/phloem, with only two papers linking them to stomata. Notably these two papers look at two classes, BRX/BRXL and PRAF. And in both, phenotypes are only observed in multiple mutants due to genetic redundancy. Of course, Arabidopsis can be different from Barley, and certainly there are examples from shoot meristem development that gene families that are redundant in one are not in the other. Still, the role of Cer-s here is explained on the back of Arabidopsis BRX family proteins, and so it is important to understand the Arabidopsis data.

- Is the methodology sound? Does the work meet the expected standards in your field? - Is there enough detail provided in the methods for the work to be reproduced?

The methods were well described, and many different analyses were used, some of which I cannot evaluate.

Reviewer #2 (Remarks to the Author):

"Converged signalling components coordinate epidermal patterning and cuticle deposition in barley" by Liu, Jose, and coauthors identifies and characterizes two genes that are required for epidermal cell fate determination and cuticular wax deposition in barley. The investigation of these two genes is thorough, covering the characterization of the biochemical, physiological, and developmental mutant phenotypes, transcriptome analysis of the mutants, investigation of sequence variation among different barley haplotypes, as well as among different plant phyla. This is an interesting manuscript, a very important contribution to the field, and it is well-written. I have only minor suggestions and questions.

The introduction to regulation of stomatal patterning is very brief, requiring good background knowledge on the part of the reader. I understand that this is the nature of such publications, however I really think it would be beneficial, if there is any room for additional text, to provide a more detailed explanation. This will make the rest of the text more interesting and accessible to a broader audience. (The background on cuticular wax metabolism is more detailed, despite the fact that it is less central to the rest of the manuscript. Perhaps the attention invested in these two topics could be shifted in the introduction).

Figure S1: Most of the features described here are related to epicuticular wax; however, toluidine blue staining reflects the integrity of the entire cuticle, not only epicuticular wax. This is a very minor point, but as there is substantial confusion about the distinction between cuticle/cuticular wax/epicuticular wax in the literature, I would suggest using "cuticle" in the title for clarity here.

HvYDA2 isn't introduced in the manuscript until the discussion. As far as I can tell, it first appears in Figure S6, and I found it a little confusing there. Could it be either briefly explained in the results, or even in the supplementary figure legend?

Figure 5 & discussion: Extant algae and land plants were studied and discussed here, so I would discourage describing any of these as "early land plants" (or "early-diverged"). I realize this is a contentious issue and one that is rapidly evolving, and so I point this out with a lot of apprehension of being a fussy reviewer. I would recommend Delaux et al., 2019, (<https://doi.org/10.1016/j.cub.2019.09.044>) and/or McDaniel et al., 2021, (<https://doi.org/10.1111/nph.17241>) for interesting and non-pedantic explanations of why these terms are misleading.

Somewhat related to this: The classification of the model moss *Physcomitrella patens* was recently re-evaluated, and it was consequently re-named *Physcomitrium patens* (<https://doi.org/10.1111/jse.12516>). Please update the nomenclature here.

Line 1022: Use of a single reference/housekeeping gene can produce misleading results, as it is rare that any given reference gene is truly consistently expressed in all tissues, developmental stages, and conditions. If possible, I would encourage the authors to quantify the expression of at least one other reference gene in their cDNA samples, to confirm a consistent result in the expression patterns they report based on use of HvACTIN7.

A couple of things were puzzling to me. Perhaps these reflect the extent of my knowledge more than anything else; I certainly don't feel this has anything to do with a shortcoming in the manuscript. I would nevertheless appreciate a response from the authors, either directly in the response letter or in their manuscript:

Is it known that stomatal spacing (and specifically the one-cell spacing rule) can be modified, within a given plant species, in order to regulate gas exchange? My understanding is that pore opening and closing, mediated by guard cell turgor, is responsible for this, and that changes in the one-cell spacing rule generally have negative consequences for plant health. I expect regulation of the one-cell spacing rule to be robust. On the other hand, it is very well-established that cuticular wax deposition is regulated in response to environmental cues, in order to control transpiration. With these assumptions, it is difficult for me to understand why the same signalling cascade that regulates wax deposition would also coordinate the one-cell spacing rule. Could the authors comment on this please? I realize that there is precedent with the *hic* Arabidopsis mutant, but with the particularly strange epidermal cell patterning in *cerg* and *cers* here, I'm genuinely curious how this can be, and suspect that the authors could share some insight.

Perhaps related: Cuticular wax composition and load is different in different epidermal cell types. Could it simply be that in the *cerg* and *cers* mutants, with their disrupted epidermal cell determination and patterning, the overall wax load and composition change as a consequence of altered development? For example, pavement cells could be producing some guard cell-like waxes, or subsidiary cell-like waxes, or perhaps silencing wax production locally because of abnormal regulation of gene expression associated with wax synthesis.

This is to an extent already suggested in the discussion, but the impression I had was that this was part of a larger framework with mutual reinforcement between cell fate and cuticle deposition. I would really be interested in knowing whether the authors consider it a possibility that the cuticular phenotypes of the *cerg* and *cers* mutants are simply downstream effects of impaired development.

Reviewer #3 (Remarks to the Author):

Cuticle is a protecting film covering the epidermis of plant shoot, which consists of lipid and hydrocarbon polymers impregnated with wax. Stomata is a pore in the epidermis of plant leaves, stems and other organs. Both cuticle and stomata play critical roles for plants in dealing with the biotic and abiotic stresses. In this manuscript, the authors identified the cer-g and cer-s through map-based cloning as YDA and BRX, respectively, which were previously reported to act in concert to regulate epidermal patterning. In addition, the authors explored the haplotypes of these two genes in a barley population and their potential roles in plant evolution. The manuscript is very well written and the subject which revealed the link between cuticle formation and epidermal patterning is really interesting. However, to publish this finding in NC journal, the authors must provide much more evidences. Some suggestions were listed below hopefully to help authors make the story more systematic and perfect.

1) Though the efficiency for barley genetic transformation is still not high, the authors had better use the CRISPR tool to generate new mutation version to verify that the two loci are indeed YDA and BRX. Alternatively, screening new mutated alleles from a TLLING pool will be useful.

2) L419, the authors claimed "cuticle properties could impact epidermal cell fate asymmetry". Is it literaturely supported? In my view, if other cuticle-related mutants do not show differences in cell fate determinacy, the conclusion might be wrong.

3) L399, as mentioned by the authors, since the grasses lack BASL homologues, HvYDA1 may interact with HvBRXL1 directly or via other intermediates. It is a very interesting point to be further confirmed. It is not complex or time consuming and I suggest the authors test whether the two components interact directly in barley.

4) L340-L341, "green algae" repeatedly appeared.

5) By constructing a phylogenetic tree and analysing the motifs of YDA and BRX orthologues, the authors identified the land plant-specific motifs, and suggested them an important role in plant terrestrialisation. It is a quite interesting point if there are functional divergence among the different versions of YDA and BRX. Some experiments, such as introducing them into Arabidopsis mutants, will be useful to test this hypothesis.

6) Similarly, the authors analysed the haplotypes of the YDA and BRX, and found some very interesting points, e.g. the potential correlation between HP_25 and hot-dry adaption. However, they are not experimentally supported.

7) Finally and most importantly, the essential information of this study is finding the link of two critical phenotypes via YDA-BRX module. The next step, the authors should clarify where this main axis branched, that is, identify the downstream components of YDA-BRX which regulates epidermal patterning and cuticle, respectively. Then constructing a more comprehensive and accurate network.

The authors must answer the last question and should try their best to solve the other ones.

Responses to Reviewer Comments

We are grateful to all three reviewers for their considered and helpful comments on our initial manuscript. In all cases where possible, we have addressed their concerns. Please see the text below for a point by point response.

Reviewer #1

Reviewer #1's major experimental concern about our identification of the gene underlying the *Cer-s* locus.

"1) Major experimental concern. I am not entirely convinced that the HvBRX is the causal gene for *cer-s*, or at least that the developmental phenotypes are caused by the lack of the same gene as the wax phenotypes. The reasons for this are elucidated below, but I think this data really needs to be confirmed before anything else can be interpreted.

.... The mapping of these genes is a heroic task, but there is little validation of gene identity. I do not expect that there would be complementation or functional studies in Barley, but there needs to be some more convincing data that causal genes for *cer-g* and *cer-s* are indeed HvYDA and HvBRXL1, and that the stomatal and wax phenotypes are due to disruption of the same gene. As far as I can tell from Sup table 2, only two *cer-g* and one *cer-s* line have stomatal clustering defects. Many lines with mutations disrupting the loci were only measured for wax.

.... To me, without a solid foundation that *cer-s* is HvBRXL1 and this locus is responsible for the epidermal phenotypes, the rest of the work is hard to interpret. But I do think there are some fairly straightforward ways to fix this....

quantify the epidermal phenotypes in at least 3 of the with nonsense or frameshift mutation in HvBRXL from among the list in Table S2. Doing the same with an additional line of HvYDA with a nonsense mutation (rather than 3 bp change that affects two amino acids in the kinase domain) would make it somewhat clearer that the *cer-g* phenotype was due to loss, rather than reduction of HvYDA function."

Author response: We have stomatal patterning data for all *cer-g* and *cer-s* alleles which support our gene identifications. These data were published in the PhD thesis of co-author S. Jose (Jose, 2016, University of Bristol) and were part of Table S2; however, regrettably these data were inadvertently omitted when we reformatted tables during the last steps of manuscript preparation. We apologise for this error. We now include these data, along with new data for stomata, prickle hair and silica cells phenotyping, as well as epicuticular wax scoring, of selected frame-shift or deletion *cer-s* and *cer-g* alleles, chosen as recommended by the reviewer (lines 285-288; Table S2). These data conclusively show that all *cer-s* and *cer-g* alleles with mutations in their respective genes share all these phenotypes, demonstrating that the causal genes are correct.

Reviewer #1 also asked for more information about the double mutant validation and commented on the analyses of double mutant phenotypes:

"I read the description of mapping and the creation of the double *cer-g cer-s*. As much as I love the classical genetics used here, as far as I can tell the double mutant was never confirmed in terms of molecular defects in the HvYDA and HvBRXL1 loci.

In Figure 2, the phenotypes of the double *cer-g cer-s* mutant sometimes looked additive and sometimes the *cer-g* phenotype predominates. These results are hard to reconcile with a model of their gene products working as a complex to guide both epidermal fate and wax deposition. In fact, what they really look like to me is that the epidermal fate and proliferation phenotypes of *cer-s* are due to another mutation loosely linked, and in the crosses, they have lost this second mutation.”

Author response: The genetic identity was unknown at the time when we conducted the stomatal and wax phenotyping on the double mutant as presented in the original manuscript. We agree with the reviewer that we should have molecularly confirmed the double mutant stock and we thank the reviewer for highlighting this oversight. To do this, we grew, and sequenced multiple individual plants grown from the double mutant stock. We discovered that the double mutant stock was not correct, thus clarifying some of the hard to reconcile data noticed by Reviewer 1. We took this opportunity to phenotype and genotype a large collection of different combinations of mutant alleles. We discovered that the *cer-g.10 cer-s.31* double mutant showed equivalent to more severe prickly hair and silica cell clustering phenotypes as *cer-s.31*, different from the weaker *cer-g.10* phenotypes: i.e. for these traits, *cer-s.31* is epistatic to and sometimes synergistic with *cer-g.10*. Furthermore, we found that a single copy of the *cer-s.31* allele is an epistatic modifier of the homozygous *cer-g.10/cer-g.10* phenotype, often converting it to close to a *cer-s.31* mutant phenotype or intermediate between *cer-s.31* and *cer-g.10* (*cer-s/+ cer-g/+* heterozygotes showed no phenotype). Taken together, these data support genetic epistasis of *Cer-s* over *Cer-g*. We have removed original double mutant phenotyping from our revised manuscript and now include a separate phenotyping and genotyping double mutants section (lines 294-311; Figure 3a-d; Table S6; Supplemental Figure 10). We thank the reviewer again for their suggestion and feel that our new data improve our paper and strengthen our hypothesis that these two genes interact.

Reviewer #1’s major writing concern was that our manuscript was overly speculative.

“The ratio of data to discussion and speculation is tilted quite far in the direction of speculation. The speculation concerns the mode of action of the genes underlying *cer-g* and *cer-s*, the effect of mutations in *cer-g* on plant adaptation, and the role of these genes in evolution of the cuticle and life on land.

Some of the ideas reported in the results should really be in the discussion (and the speculation toned down considerably).

For example, lines 190-193, “Collectively our data reveal that HvYDA1 and HvBRXL1, orthologues of genes which work together to control stomatal development in Arabidopsis, likely work together to promote cuticular integrity and wax deposition in barley”. I disagree with the single BRX-domain protein being called an orthologue, that there is ANY evidence that BRXL genes and YDA work together as a complex in Arabidopsis, and am skeptical of the claim that the two work TOGETHER in barley.

Likewise, there is no actual evidence for “shared pathways” (line 253, line 275), nor are the clear data showing a “molecular insight into how HvYDA1 and HvBRXL1 regulate multiple features during epidermal patterning (line 301).

It is a somewhat smaller point, but the naming of the gene that contains a single BRX domain as BRXL1 will cause confusion in the field. In part, because the term BRXL already is used for 2 BRX-domain proteins for at least a decade (Hardtke lab papers) and this nomenclature is continued in

other labs. BRX domains are present in other proteins as well, such as the PRAFs (recently described in Wang et al., 2022, Nat Comm). The mechanisms by which PRAFs and BRX family proteins work are very different and without any experimental data showing functions or interactions of the Barley single BRX-domain with other proteins (such as YDA), the idea of a YDA/HvBRX complex needs to be toned down. “

Author response: We understand the reviewer’s concern and have toned down and qualified our text concerning the mechanism used by the genes underlying *Cer-g* and *Cer-s* (lines 457-459) and possible roles of *cer-g* haplotypes in crop responses to stress (line 376 and 536). We appreciate the reviewer’s concern about describing the *Cer-s* gene as a *BRXL1* orthologue given the difference in domain structure. In our revised manuscript, we do not describe *HvBRXL1* as orthologous and instead rename *HvBRXL1* as *HvBRX-Solo* (line 198) to reflect the solitary BRX domain structure. We believe that including BRX in the gene name is reasonable and will be acceptable to the community; for instance, the Arabidopsis gene containing a single BRX domain is annotated as a *BRXL* in TAIR. We now refer to shared pathways only in the context of our genetic analyses which we feel provide a strong support for genetic interaction and a solid rationale for future work to explore the function of HvYDA1 and HvBRX-Solo and any intermediary partners.

Sections about the adaptive significant of HvYDA alleles in natural populations should be moved out of the results section as there are no functional or pop-gene analyses of these alleles. ”

Author response: We have removed this text to the discussion (lines 534-539).

Reviewer #1’s suggestions:

“Figure 3 ... title: Variation in HvYDA1 and HvBRXL1 causes similar expression changes in genes involved in epidermal development does not seem to fit the data in e. where the magnitude of change genes in *cer-g* and *cer-s* is quite different. The gene expression appears to be a perfect match for the strength of the phenotype, and I don’t have any concerns with the data itself. I do think it is quite weak evidence that the genes underlying *cer-g* and *cer-s* act in a common pathway. Two completely independent pathways that resulted in wax and stomatal phenotypes would give the same transcriptional results.

Author response: We have modified our title to “HvYDA1 and HvBRX-Solo regulate a cuticle and epidermal patterning enriched transcriptome” (line 345) in line with these suggestions.

Figure 4. As a representation of allele distribution this is fine. The suggestion that mutations affecting phosphorylation sites have any meaning is dubious.

Author response: We are puzzled by the reviewer comment that suggesting functional relevance to mutations affecting phosphorylation sites in kinases is ‘dubious.’ Multiple studies show that MAPKKKs activity is regulated by addition and removal of phosphates onto serine/threonine residues outside of the catalytic domain, including Ste11-like MAPKKKs (eg. van Drogen et al 2000 Current Biol). In fact, studies in Arabidopsis showed that AtYDA1 is a target of phosphorylation by the GSK3 kinase BIN1 whose activity was important to regulate AtYDA1 function (Kim et al., 2012 Nature). Several putative GSK3 phosphorylation sites (Ser/Thr-x-x-x-Ser/Thr), including those highlighted in Kim et al (2012) show non-synonymous changes in HvYDA1 haplotype 25, and are highly conserved across the YDA clade, suggestive of a conserved and important function. While we do not provide direct evidence that these sites regulate HvYDA1 activity, we believe that conservation with Arabidopsis YDA is plausible enough to mention (lines 361-363). We 100% agree with the reviewer that these are assumptions only which must be tested (and we are developing germplasm as part of a longer-term project to do this), and we have reworded to emphasise this point (lines 536-540).

Figure 6. HvYDA1 and HvBRX are shown as if acting as a complex. But there is no evidence in any species that they would. Even taking the best characterized systems in Arabidopsis, BRX does not interact with MAPKKs, but rather with D6PKs, and BASL, a protein that is not outside of angiosperms, is the glue between these proteins. A complex of the two proteins working together exclusively is inconsistent with the divergent phenotypes in the cer-g and cer-s lines and reported epistatic effect of cer-g.

Author response: We regret that the diagram showed a HvYDA1/HvBRXL1 complex – we intended this drawing to show that these factors shared roles only. We agree with the reviewer that there is no evidence that these proteins directly interact and that other proteins such as BASL are important to link these proteins together in certain species. We have modified our model (Fig 7) accordingly.

- Are there any flaws in the data analysis, interpretation and conclusions? - Do these prohibit publication or require revision?

Many of the interpretations go quite a bit beyond the data and, in particular with HvBRX, make a fair bit of assumption. The BRX genes were also quite well studied in roots/phloem, with only two papers linking them to stomata. Notably these two papers look at two classes, BRX/BRXL and PRAF. And in both, phenotypes are only observed in multiple mutants due to genetic redundancy. Of course, Arabidopsis can be different from Barley, and certainly there are examples from shoot meristem development that gene families that are redundant in one are not in the other. Still, the role of Cer-s here is explained on the back of Arabidopsis BRX family proteins, and so it is important to understand the Arabidopsis data.

Author response: We are fascinated by the non-redundant role for the single gene *HvBRX-Solo* in barley epidermal patterning. We are also curious as to the level of redundancy which could exist in the control of these and other traits. We hope to mobilise and/or generate new germplasm to explore this possibility in future work. We now include a statement contrasting the redundant functions of other BRX-domain and PRAF proteins in Arabidopsis which the role for *HvBRX-Solo* (lines 460-461).

- Is the methodology sound? Does the work meet the expected standards in your field? - Is there enough detail provided in the methods for the work to be reproduced?

The methods were well described, and many different analyses were used, some of which I cannot evaluate.

Reviewer #2:

We thank Reviewer #2 for many helpful and supportive comments about our manuscript. We are very happy to address Reviewer #2's minor suggestions and questions below in a point by point response.

"...I really think it would be beneficial, if there is any room for additional text, to provide a more detailed explanation [of stomatal patterning]... The background on cuticular wax metabolism is more detailed, despite the fact that it is less central to the rest of the manuscript. Perhaps the attention

invested in these two topics could be shifted in the introduction”

Author response: We now provide more details about stomatal patterning and recent advances in Arabidopsis and grasses in our revised manuscript (lines 86-105).

Figure S1: Most of the features described here are related to epicuticular wax; however, toluidine blue staining reflects the integrity of the entire cuticle, not only epicuticular wax. This is a very minor point, but as there is substantial confusion about the distinction between cuticle/cuticular wax/epicuticular wax in the literature, I would suggest using “cuticle” in the title for clarity here.

Author response: We agree with the reviewer and have modified the title as suggested.

HvYDA2 isn’t introduced in the manuscript until the discussion. As far as I can tell, it first appears in Figure S6, and I found it a little confusing there. Could it be either briefly explained in the results, or even in the supplementary figure legend?

Author response: We now introduce HvYDA2 and its expression earlier in the results section (lines 209-212)

Figure 5 & discussion: Extant algae and land plants were studied and discussed here, so I would discourage describing any of these as “early land plants” (or “early-diverged”). I realize this is a contentious issue and one that is rapidly evolving, and so I point this out with a lot of apprehension of being a fussy reviewer. I would recommend Delaux et al., 2019, (<https://doi.org/10.1016/j.cub.2019.09.044>) and/or McDaniel et al., 2021, (<https://doi.org/10.1111/nph.17241>) for interesting and non-pedantic explanations of why these terms are misleading.

Somewhat related to this: The classification of the model moss Physcomitrella patens was recently re-evaluated, and it was consequently re-named Physcomitrium patens (<https://doi.org/10.1111/jse.12516>). Please update the nomenclature here.

Author response: We have updated the nomenclature and have removed “early” terminology and thank the reviewer for the helpful references.

Line 1022: Use of a single reference/housekeeping gene can produce misleading results, as it is rare that any given reference gene is truly consistently expressed in all tissues, developmental stages, and conditions. If possible, I would encourage the authors to quantify the expression of at least one other reference gene in their cDNA samples, to confirm a consistent result in the expression patterns they report based on use of HvACTIN7.

Author response: We have included a second reference gene and updated the figures. The results were the same.

A couple of things were puzzling to me. Perhaps these reflect the extent of my knowledge more than anything else; I certainly don’t feel this has anything to do with a shortcoming in the manuscript. I would nevertheless appreciate a response from the authors, either directly in the response letter or in their manuscript:

Is it known that stomatal spacing (and specifically the one-cell spacing rule) can be modified, within a given plant species, in order to regulate gas exchange? My understanding is that pore opening and closing, mediated by guard cell turgor, is responsible for this, and that changes in the one-cell spacing rule generally have negative consequences for plant health. I expect regulation of the one-cell spacing rule to be robust. On the other hand, it is very well-established that cuticular wax deposition is regulated in response to environmental cues, in order to control transpiration. With

these assumptions, it is difficult for me to understand why the same signalling cascade that regulates wax deposition would also coordinate the one-cell spacing rule. Could the authors comment on this please? I realize that there is precedent with the *hic* Arabidopsis mutant, but with the particularly strange epidermal cell patterning in *cerg* and *cers* here, I'm genuinely curious how this can be, and suspect that the authors could share some insight.

Perhaps related: Cuticular wax composition and load is different in different epidermal cell types. Could it simply be that in the *cerg* and *cers* mutants, with their disrupted epidermal cell determination and patterning, the overall wax load and composition change as a consequence of altered development? For example, pavement cells could be producing some guard cell-like waxes, or subsidiary cell-like waxes, or perhaps silencing wax production locally because of abnormal regulation of gene expression associated with wax synthesis.

This is to an extent already suggested in the discussion, but the impression I had was that this was part of a larger framework with mutual reinforcement between cell fate and cuticle deposition. I would really be interested in knowing whether the authors consider it a possibility that the cuticular phenotypes of the *cerg* and *cers* mutants are simply downstream effects of impaired development.

Author response: Several papers describe stomatal clusters in wild species (eg. Metcalfe and Chalk, 1979, found stomatal clusters in more than 60 species), with some papers linking these phenotypes with improved WUE (eg. Papanatsiou et al. 2007 [10.1093/jxb/erx072](https://doi.org/10.1093/jxb/erx072)) and others showing changes in stomatal clustering in response to environmental variation. For instance, Zhao et al (2006, <https://doi.org/10.1016/j.sajb.2006.03.006>) found a negative correlation between stomatal clustering in leaves of *Cinnamomum camphora* (camphor tree) and soil moisture, which could reflect a role in water conservation. In another study, *Vicia faba* L. increased contiguous stomatal clustering in step with increased drought and salinity stress, suggesting that plants can induce/increase stomatal clustering in adaptive responses to environmental cues (Gan et al, 2010, Botanical Studies 51: 325-336). We now mention these studies in our discussion (lines 528-531). The molecular mechanism underlying this plasticity in stomatal spacing is unknown. Given the role of MAP kinase cascades in environmental responses, the YDA-driven pathway could be involved. In our study, we found haplotypes exclusive to the cultivated germplasm which show changes in putative GSK3 phosphorylation sites on HvYDA1 – conserved sites thought to regulate YDA function in Arabidopsis. We plan to use these germplasm in future work to understand whether YDA sequence variation correlates with variation in stomatal spacing.

Like the reviewer, we are intrigued that a signalling pathway which ensures stomatal spacing, and accordingly stomatal efficiency, also controls cuticular integrity and other epicuticular specialisations important for resiliency in terrestrial environments. The wax bloom responds to environmental cues, likely exposure to the atmosphere, but only occurs on reproductive stage tissues (leaf sheaths, elongated internodes and spikes), suggesting a role for developmental stage in epidermal differentiation. For instance, we note in our discussion that silica cells are only present on reproductive stage tissues and that these cells are thought to originate some of the wax deposition. Trichome patterning is linked to vegetative phase change in Arabidopsis, while in maize Glossy15 encodes an miR172-regulated AP2 which “co-ordinately activates the expression of cell-specific juvenile epidermal traits (waxes and cell wall characteristics) and suppresses the differentiation of adult epidermal cell types (e.g., bulliform cells and epidermal hairs)” (Moose and Sisco, 1995). However, the same study showed that “the epidermal tissue of transition nodes often simultaneously expresses both juvenile biochemical traits ...and an adult pattern of differentiation, ...suggesting that these two epidermal phenotypes are not mutually exclusive cell fates and are regulated independently.” Furthermore, while these traits are coordinated by GL15 they appear independent from each other and cell autonomous based on sector analysis from GL15 activation in different tissue layers. These results agree with Freeling and Lane (1994) and Sturaro et al (2005) showing that epicuticular wax production is cell autonomous. So, HvYDA1 and HvBRX-Solo could

control multiple epidermal traits but independently of each other, for instance, cuticular integrity and epidermal spacing in young leaves.

On the other hand, the later deposition of epicuticular wax bloom occurs long after epidermal fate decision. Our revised manuscript describes our new gene expression data showing that *HvYDA1* and *HvBRX-Solo* is almost exclusively expressed in basal leaf sheath sections (during epidermal differentiation) but not in older tissue prior to emergence which expresses the *Cer-CQU* gene cluster important for beta-diketone biosynthesis in emerging tissues (lines 212-222). These data support that the role of *HvYDA1* and *HvBRX-Solo* in epicuticular wax bloom may reflect an earlier function, perhaps those involved in epidermal cell fate. We explore these ideas in our discussion (lines 505-518).

Reviewer #3 (Remarks to the Author):

Cuticle is a protecting film covering the epidermis of plant shoot, which consists of lipid and hydrocarbon polymers impregnated with wax. Stomata is a pore in the epidermis of plant leaves, stems and other organs. Both cuticle and stomata play critical roles for plants in dealing with the biotic and abiotic stresses. In this manuscript, the authors identified the *cer-g* and *cer-s* through map-based cloning as *YDA* and *BRX*, respectively, which were previously reported to act in concert to regulate epidermal patterning. In addition, the authors explored the haplotypes of these two genes in a barley population and their potential roles in plant evolution. The manuscript is very well written and the subject which revealed the link between cuticle formation and epidermal patterning is really interesting. However, to publish this finding in NC journal, the authors must provide much more evidences. Some suggestions were listed below hopefully to help authors make the story more systematic and perfect.

1) Though the efficiency for barley genetic transformation is still not high, the authors had better use the CRISPR tool to generate new mutation version to verify that the two loci are indeed *YDA* and *BRX*. Alternatively, screening new mutated alleles from a TLLING pool will be useful.

Author response: Our extensive, independent allelic series for both *HvYDA1* and *HvBRX-Solo* conclusively demonstrates that these two genes underlie the *Cer-g* and *Cer-s* loci. We feel that gene editing, beyond taking about 2 years to complete, would only serve to make another allele and not advance our understanding beyond the current manuscript.

2) L419, the authors claimed “cuticle properties could impact epidermal cell fate asymmetry”. Is it literately supported? In my view, if other cuticle-related mutants do not show differences in cell fate determinacy, the conclusion might be wrong.

Author response: There is long standing evidence that cuticular mutants also show epidermal cell fate differences. As mentioned in our manuscript introduction (lines 124-129), overexpression of the genes encoding SHINE1/WIN1 transcription factors in Arabidopsis influence trichome and stomatal development in addition to cuticular wax deposition. More recently, a major new paper published in Plant Cell connect cuticular features with stomatal fate in Arabidopsis (Yang et al., 2022). Our introduction and discussion sections discuss previous research linking these two traits (lines 124-132; lines 469-502). So our findings are well placed in the literature.

3) L399, as mentioned by the authors, since the grasses lack BASL homologues, *HvYDA1* may interact with *HvBRXL1* directly or via other intermediates. It is a very interesting point to be further confirmed. It is not complex or time consuming and I suggest the authors test whether the two components interact directly in barley.

Author response: We also agree that this is an interesting avenue to explore. However, we content that these experiments are neither simple nor fast in barley. Making stable transgenics takes years, at high cost and laborious. Transient expression in barley is not widely published or routine. We feel these experiments are best suited for another paper examining the molecular mechanisms of YDA1 and BRX-Solo in barley.

4)L340-L341, “green algae” repeatedly appeared.

Author response: fixed

5) By constructing a phylogenetic tree and analysing the motifs of YDA and BRX orthologues, the authors identified the land plant-specific motifs, and suggested them an important role in plant terrestrialisation. It is a quite interesting point if there are functional divergence among the different versions of YDA and BRX. Some experiments, sun as introducing them into Arabidopsis mutants, will be useful to test this hypothesis.

Author response: We agree with the reviewer that these experiments may reveal interesting functional differences between species. However, we do not feel this experiment is essential for our findings presented in this paper and would rather include this sort of investigation in another paper comparing mechanistic divergence. We also point out that papers studying these factors in Arabidopsis do not extend their examination into monocot models to test for functional divergence.

6)Similarly, the authors analysed the haplotypes of the YDA and BRX, and found some very interesting points, e.g. the potential correlation between HP_25 and hot-dry adaption. However, they are not experimentally supported.

Author response: We 100% agree that this is a correlation only and that our manuscript does not provide experimental evidence that these haplotypes confer adaptive advantages in hot and dry climates. We have rewritten the result and discussion text to emphasise this point and mention that these are hypotheses to pursue and predictions to test (lines 537-539).

7)Finally and most importantly, the essential information of this study is finding the link of two critical phenotypes via YDA-BRX module. The next step, the authors should clarify where this main axis branched, that is, identify the downstream components of YDA-BRX which regulates epidermal patterning and cuticle, respectively. Then constructing a more comprehensive and accurate network.

The authors must answer the last question and should try their best to solve the other ones.

Author response: We feel this is not yet understood for Arabidopsis, let alone barley. These experiments represent another research paper examining the downstream mechanics of this pathway.

REVIEWERS' COMMENTS

Reviewer #1 (Remarks to the Author):

The major issues that were required for publication have been largely remedied in this revision. The discussion points about terrestrialization and the evolution of YDA and BRX-domain proteins vis a vis stomata are still made with quite a bit more certainty than the data warrant. It would not weaken this story at all to cut back on the excessive speculation and allow the strong experimental and genetic data to shine.

Some smaller issues are:

Ln 131 com mon should be common

Ln260, high [CO₂] experiments. The high [CO₂] increased stomatal density of WT to match that of the mutants, which could mean that they are insensitive, or that the mutants are already at the top of the range and the environmental effect is hidden. The data appear quite noisy and the effect is quite modest and this part of the manuscript could be removed without major impact on the main story.

Ln527, I don't see any evidence that production of contiguous stomata is the trait that results in increased drought and salinity stress, rather than this comes along as a not-so-terrible secondary effect of another morphological adaptation. I'd also be careful about using ref78 to bolster an argument about contiguous stomata as most "clustered" Begonia stomata are not actually in contact, likewise ref 79 takes the more cautious interpretation that contiguous stomata are "markers" of environmental stress, not an adaptation to protect against it.

Reviewer #2 (Remarks to the Author):

"Converged signalling components coordinate epidermal patterning and cuticle deposition in barley" by Liu, Jose, and coauthors identifies and describes CER-G and CER-S that are required for epidermal cell fate determination and normal cuticular wax deposition in barley. The investigation of these two genes includes the characterization of the biochemical, physiological, and developmental mutant phenotypes, transcriptome analysis of the mutants, investigation of sequence variation among different barley haplotypes, as well as among different plant phyla and green algae. This is a fascinating manuscript, it is an important contribution to the field, it is very well-written, and the authors have taken every possible step to address my comments as well as those of the two other reviewers. There were many substantial improvements to the manuscript in this second submission, including a thorough investigation of the double mutant as suggested by reviewer 1, two new and exciting sections in the discussion in response to my and reviewer 3's questions, and gene expression profiling in different maturation stages of leaf sheaths. I do not have any remaining critical feedback for the authors and congratulate them on this fantastic discovery and manuscript.

Reviewer #3 (Remarks to the Author):

This study is important because it uncovers a link between cuticle formation and epidermal patterning. The authors have now adjusted some clarifications and revised them in the new version as needed. I completely understand the difficulties of barley material generation and the author's feelings. If the same study were done in maize or rice, extensive genetics would have to be required to reach the level of NC journal.

We thank all three reviewers for their positive reception of our revision. Reviewer #1 suggested further changes. We address these comments below:

Reviewer #1 (Remarks to the Author):

The major issues that were required for publication have been largely remedied in this revision. The discussion points about terrestrialization and the evolution of YDA and BRX-domain proteins vis a vis stomata are still made with quite a bit more certainty than the data warrant. It would not weaken this story at all to cut back on the excessive speculation and allow the strong experimental and genetic data to shine.

Author response: We rewrote the last paragraph of our discussion, removing more speculative statements and links between our discovery of a putative pathway controlling multiple epidermal specialisations and a potential role in terrestrial adaptation (lines 522-547).

Some smaller issues are:

Ln 131 com mon should be common

Author response: fixed.

Ln260, high [CO₂] experiments. The high [CO₂] increased stomatal density of WT to match that of the mutants, which could mean that they are insensitive, or that the mutants are already at the top of the range and the environmental effect is hidden. The data appear quite noisy and the effect is quite modest and this part of the manuscript could be removed without major impact on the main story.

Author response: We removed this data panel.

Ln527, I don't see any evidence that production of contiguous stomata is the trait that results in increased drought and salinity stress, rather than this comes along as a not-so-terrible secondary effect of another morphological adaptation. I'd also be careful about using ref78 to bolster an argument about contiguous stomata as most "clustered" Begonia stomata are not actually in contact, likewise ref 79 takes the more cautious interpretation that contiguous stomata are "markers" of environmental stress, not an adaptation to protect against it.

Author response: We rewrote this section to considerably tone down any link between stomatal clustering and adaptation or improved environmental tolerance (lines 522-535).